# Death of backcountry winter-sports practitioners in avalanches – A systematic review and meta-analysis of proportion of causes of avalanche death

**Guang Rong**[1,2]☯, **Lauri Ahonen**[1,2]☯, **Gerit Pfuhl**[2,3], **Benjamin Ultan Cowley**[1,2,4]¤*

**1** Faculty of Educational Sciences, University of Helsinki, Helsinki, Finland, **2** Center for Avalanche Research and Education, UiT the Arctic University of Norway, Tromsø, Norway, **3** Department of Psychology, Norwegian University of Science and Technology, Trondheim, Norway, **4** Cognitive Science, Faculty of Arts, University of Helsinki, Helsinki, Finland

☯ These authors contributed equally to this work.
¤ Current address: University of Helsinki, Helsinki, Finland
* ben.cowley@helsinki.fi

**Data availability statement:** Data and code is archived at OSF (https://osf.io/ygza4/)

## Abstract

This study estimates the proportions of the three major causes of avalanche death globally, and reviews potential factors influencing the proportions of causes of avalanche-related deaths (PCAD). By searching databases and consulting experts, we retrieved studies and registries in multiple languages, which examined PCAD. As a result, we retrieved 1,415 reports and included 37 for the study (22 for meta-analysis). We performed a meta-analysis to estimate pooled proportions. Between-study heterogeneity was assessed jointly by $I^2$ and 95% prediction interval of pooled estimates. PCAD by trauma and asphyxia are 29% (95%CI 21–39%) and 82% (95%CI 72–88%), after the year of 2000. PCAD by hypothermia is 2% (95%CI 1–4%), estimated with studies having sufficient sample size. Time periods (before or after 2000), data representativeness (national subgroup), forensic procedures, and sample size explained between-study variation for proportions to a considerable extent. Factors influencing PCAD, that were either available or not available for quantitative synthesis, were summarized in a narrative systematic review (37 studies). In conclusion, we re-affirm asphyxia as the predominant cause of avalanche death, followed by trauma, and then hypothermia. Patterns of PCAD by trauma and asphyxia varied more after the year of 2000. A sample size > 75 is needed to estimate the proportion of hypothermia. PCAD discrepancies are lower in the data representing fatalities from a country than from regions. Without proper forensic diagnosis procedure, PCAD by trauma can be over-estimated. Under-reporting of forensic diagnostic criteria is an important bottleneck to the reliability of evidence in the field. Evidence on the role of other influencing factors to PCAD such as fatalities' expertise and usage of mitigation gear is anecdotal and warrants further research. The results of meta-analysis build upon synthesizing and summarizing studies with moderate to high risk of bias and should be interpreted with caution.

**Funding:** This study was financially supported by NordForsk, as part of the "Grappling with Uncertainty in Environments Signaling Spurious Experiential Decisions (GUESSED)" project, in the form of a grant (105061) received by GR, LA, GP and BUC. This study was also financially supported by UiT The Arctic University of Norway in the form of an award. This study was also financially supported by the University of Helsinki in the form of salaries for GR and LA. The specific roles of these authors are articulated in the 'author contributions' section. No additional external funding was received for this study. The funders had no role in study design, data collection and analysis, decision to publish, or preparation of the manuscript.

## Introduction

In the Alpine countries, avalanche disasters have been mentioned in the literature as long ago as 218 BC [1]. Our long-standing cultural familiarity with these destructive phenomena belies the fact that their nature and consequences are still not fully understood. An avalanche occurs on a slope when the cohesive properties within the snow-pack or between the snow and the ground fail to resist downward gravitational forces [2–4]. With modern avalanche control in place to protect businesses and communities, the majority of people facing avalanche danger are winter sport recreationists, such as skiers, snowboarders, or hikers [5,6]. These activities are associated with health benefits but also expose practitioners to risk of injury or death [7], in increasing numbers as the activities grow increasingly popular [8]. For example, according to the Colorado Avalanche Information Center, all avalanche fatalities in Colorado from the 2019–2020 season to the 2023–2024 season were related to winter sports [9]. Here, we aim to advance our understanding of the causes of death from avalanches, and how they are reported in literature.

Getting caught by an avalanche is a high-consequence event. Mortality is 20% to 70%, depending on the completeness of the burial and on the timeliness of rescue [10]. Several different mechanisms of death may occur, including mechanical trauma, hypothermia, and suffocation/asphyxia [11], and is essential to evaluate in forensic autopsy. For avalanche victims, pathologists often consider burial duration, presence of injuries, core temperature, serum potassium levels, and/or airway patency to perform forensic differential diagnoses [12–14].

Although these pathologies (asphyxia, trauma hypothermia) have a stable proportion in the general population, avalanche accident scenarios can greatly deviate from this pattern [13,15,16]. Asphyxia, defined as a condition where cells are deprived of oxygen, is pivotal to pathology research despite being relatively rare in general forensic casework [17]. In most cases, an avalanche accident involves snow burial, which either leads victims to re-breath expired gases [18], or mechanical injury to the victims' respiratory system (e.g. severe rib fracture) that degrades their ability to breathe. This makes asphyxiation a typical pathology in avalanche accidents [19]. On the other hand, the prolonged exposure to cold environment makes hypothermia (i.e., a drop in core body temperature to <35°C) [20] a more life-threatening concern for avalanche victims, compared with accidents in temperate climate zones [16]. Meanwhile, although accidental trauma, such as injuries caused by direct physical force, constitute a great proportion of cases requiring forensic examinations [13], evidence shows it is the least [21] or second-to-least [22] frequent cause of avalanche death among the three.

Different causes of avalanche deaths can be prevented with different procedures [23,24] and can lead towards different resuscitation pathways/resources such as cardiopulmonary bypass resuscitation for hypothermia [12,25]. Clarifying these patterns of proportions of causes of avalanche death (PCAD) is important in reducing mortality. Asphyxia is consistently reported as the main cause of death, whereas the proportions for the two other mechanisms are inconsistently reported. For example, Boyd *et al.* [26] found a high rate of overall trauma mortality (24%) and even higher rates among the subgroups of helicopter skiers (30%) and off-piste skiers (33%). These findings are in contrast with low mortality rates from trauma in, for example, Austria (5.6%) [27] and Utah, USA (5.4%) [18]. This could be because avalanche statistics often rely on a relatively small number of events [28]. Moreover, there could be major confounders, including but not limited to: variation in the back-country topography throughout the world, or developments in the risk-reduction measures and measures to evaluate cause of death.

Previous review studies on the mechanisms for avalanche death have had several limitations in scope. Most reviews on this topic focused on discrepancies in the existing evidence, and none examined the pooled proportions, or compared the differential effects on the outcome of such factors as, e.g. geographical region. However, indirect evidence on *mechanisms* of avalanche survival suggests that we are very likely to find a consistent global *distribution* of avalanche death causes. That is, based on the estimation of the avalanche survival probability curves first described by De Quervain in 1966 [29] (see S1 Fig), the survival probabilities of avalanche victims have four distinct phases in the survival curve. A similar avalanche survival pattern was replicated by Markwalder with Swiss avalanche accident data in 1970 [30], by Burtscher and Martin using Austrian data in 1994 [31], and by Haegeli *et al.* [32] for Canadian fatalities in 2011. Recent evidence has revealed that traumatic injuries were the main cause of death in the first phase; asphyxia in the second phase; and a combination of hypoxia, hypercapnia, and hypothermia in the third phase [33]. These studies suggest a stable pattern of causes of death distribution, which has yet to be fully elucidated by meta-analysis.

This study attempts to aggregate the current evidence by structured search, and robustly estimate the distributions by meta-analysis. We furthermore account for methodological confounders and topographical and regional factors by follow-up subgroup analysis, which provide insights into how PCAD estimates vary. Thereafter, we make a narrative systematic review of the studies. We aim to answer the following research questions:

(a) What are the proportions of avalanche death due to *asphyxia*, *trauma* and *hypothermia*, respectively?
(b) How do *study sample size*, *source of data*, *time periods of accidents* (Time period is defined as the range of years of data covered by a study), *geographic region*, or *forensic diagnostic approach* confound the PCAD estimates?
(c) What is the current status of evidence for factors influencing PCAD?

## Materials and methods

The research consists of a systematic review and a meta-analysis (Protocol registered: INPLASY202430011). The systematic review follows the reporting guidelines and criteria set in Preferred Reporting Items for Systematic Reviews and Meta-Analyses (PRISMA 2020). For more details, see S1 Table.

### Search strategy to identify studies

We defined avalanche fatalities as people who died due to being caught by an avalanche and carried some distance or partially/completely buried; or else who were killed in an effort to avoid an avalanche. We sought for literature which reported the number of avalanche fatalities along with forensic diagnoses of cause of avalanche death (or necessary information for computing these numbers).

We had three outcomes including proportions of avalanche death by trauma (1), asphyxia (2) and hypothermia (3). For this purpose, we conducted a comprehensive literature search of the following electronic databases: MEDLINE (1950 to March 2024), Academic Search Complete (1948 to March 2024), SPORTDiscus (1982 to March 2024), Embase (1980 to March 2024), ERIC (Education Resources Information Center; 1965 to March 2024), Scopus (1970 to March 2024), the Cochrane database (1993 to March 2024), and SafetyLit (1870 to March 2024). We also screened International Snow Science Workshop online databases (https://www.issw.net/) (1976 to 2023).

We designed the search strategy to be inclusive, with precision to be improved by manual screening. High recall was achieved by using a broad set of keywords (with truncation mark asterisk '*' to replace any number of characters; or question mark '?' to replace a single character), including: 'avalanche*', 'snow burial*', 'snow immers*', 'casualty', 'mortality', 'fatal*', 'death?', 'trauma', 'injur*', 'survival', 'snow*', 'ski*', 'mountain*', 'sled*', 'climb*', 'winter', 'sport*', 'activit*', 'pastime', and 'recreation'. Also, to identify all relevant English and non-English articles, no language constraints were used – thus only non-English articles with English title and abstract were returned by this search.

Specific search terms were also confined to MeSH terms or Emtree in databases providing such functions. The search results were then refined by Boolean logic combinations of the terms. For details on search strategy for each database, see S2 Table.

## Eligibility criteria and study selection

The systematic review and meta-analysis included descriptive and analytical studies involving the distribution of death causes (pathologies) due to fatal avalanche accidents. Studies reporting fatalities with clearly confirmed pathology of death were included. To allow this study to be globally applicable, no restrictions were placed on age, gender, region, or ethnicity of the victims included in the study (regional discrepancies were examined in the subgroup analyses).

The Inclusion Criteria were: descriptive and analytical studies that report PCAD.

The Exclusion Criteria for our meta-analysis were: (a) Study selected a victim group thus biasing the PCAD (e.g., study of traumatic death excluding other causes; however, we kept any study that selected specific winter sports, because evidence shows no difference in PCAD among major winter sports [34]); (b) Study classified causes of avalanche death so that asphyxia, trauma, and hypothermia were not used nor could be inferred (e.g., study classification had no more detail than "avalanche death" or "brain death"); (c) Study reported no relevant statistics (e.g. study summarised procedures to prevent avalanche trauma but without a report of statistics of death causes); (d) Study merely duplicated relevant statistics from another included study; (e) Study reported a single accident or had sample size ≤3 (there are three common pathologies for avalanche deaths; PCAD can be tremendously biased in such cases); (f) Source and duration of data were already fully covered by an included study.

For the systematic review, we used all studies included in our meta-analysis plus those excluded by criteria (a) and/or (e). This is justified because studies excluded by (a) can be controlling groups, to which we can compare the pooled proportion estimates from meta-analysis, giving an idea of the possible risk/protective factors for avalanche death; studies excluded by (e) show how PCAD can be biased, when a sample is not big enough to allow for all three causes. Additionally, studies satisfying (a) and/or (e) are part of a whole picture of PCAD studies giving original estimates and should be considered in the systematic review. In the current report, the studies excluded for meta-analysis but included in the systematic review were referred to as *SR-only* studies (namely, systematic review only).

For English articles, two of us (R.G. and L.A.) screened the titles, and abstracts when available, of potentially relevant studies. The same reviewers independently assessed the full text based on the inclusion and exclusion criteria. Disagreements were resolved by consensus and arbitrated by consultation with a third reviewer (B.C.). For non-English articles (in German, French, Norwegian, Spanish, Japanese, and Russian), we screened by consulting researchers who are native speakers, or at least fluent, in those languages; they were also all familiar with back-country sports.

## Other sources of studies

After initial inclusion of studies from database search, we checked the reference lists of all included studies for eligibility. Further, we consulted experts from Norway, Switzerland, and Germany for other eligible registries, reports or studies, focusing on non-English sources.

## Risk of bias

We regard PCAD as a special case of prevalence study where the prevalence for each cause among fatalities was investigated. As such, two of us (R.G. and L.A.) independently assessed the risk of bias of the studies using an established tool for prevalence studies [35]. Disagreements were resolved by consensus and arbitrated by consultation with a third reviewer (B.C.). We conducted one risk-of-bias assessment for each study. Although we consider three causes of avalanche death, they all arise from a single forensic strategy within each reviewed study, and hence any potential bias would affect each study as a whole rather than individual outcomes separately.

This tool appraises the risk of bias on 11 aspects (11 items) (We deviated from the registered protocol. Instead of the 6-item evaulation tool [36] we use this 11-item evaluation tool, which covers similar aspects than the 6-item tool but includes several other aspects.) including *representative target population*, *sampling frame*, *random selection*, *non-response bias*, *direct data collection from subjects*, *case definitions*, *reliable and valid measurement*, *mode of data collection*, *length of the shortest prevalence period*, *appropriate numerators and denominators*, and *summary*. For the first 10 items, the rating is either YES (low risk) or NO (High risk) [35]. If there is insufficient information in the article to permit a judgement for a particular item, the answer should be NO (high risk) for that particular item [35]. For the last item (summary), the rating can be either low, moderate, or high risk of bias [35]. For more details, see S1 Text.

## Data extraction

Two of us (R.G. and L.A.) extracted the planned parameters from the studies, including data source, sports involved, sample size, forensic diagnosis method, time period of the source data, study design, location/region of the accidents, number of victims who died due to each cause, number of each gender, age (mean or median), proportion of complete burial, mean/median burial time, and mean/median burial depth. The data were checked for completeness and accuracy; disagreements were resolved by consensus, and if necessary, by consulting a third reviewer (B.C.). A fatality was assigned with more than one cause of death if the authors have reported so. If a study used data from different cohorts and reported PCAD separately (e.g. to compare across these cohorts), we extracted each cohort as a single entry. If a study combined these sources to reach an overall proportion for legitimate reason (e.g. to make their sample more representative of a population), we extracted this overall proportion as a single entry for the study.

If two or more studies have source and duration of data partially (not fully) covered by each other, we tried to extract the non-overlapped subsets of statistics from the studies. When such subsets were not available, we recorded them as separate entries but marked the overlap condition (for sensitivity analysis). We prioritized peer-reviewed and lower risk-of-bias studies when considering from which to keep full statistics and which to extract subset statistics among overlapping studies.

### Dealing with missing data

We excluded meta-analysis studies with missing values for sample size or all of the three causes of death, if the missing data could not be procured by contacting the corresponding authors. When sample size was reported and at least one cause(s) was not missing, we included it for the meta-analysis of the non-missing cause(s) alone, even though the missing data was not procured from the authors. For SR-only studies, we included them despite any missing values, given the focus on a qualitative review.

When a study used a partially-different death cause categorization than ours (e.g. a categorization strategy of asphyxia and brain death), or when the authors indicated the causes were unknown/lost, we did not seek to procure the missing proportion for the relevant causes.

### Data synthesis for meta-analysis

Studies with extremely small sample size ($n \leq 3$) or biased sample would be misleading and consequently excluded from meta-analysis. However, they were included in the narrative systematic review, SR-only.

A generalized linear mixed model (GLMM) was used to estimate the pooled proportions for each cause of avalanche death, separately; this approach has been suggested for meta-analyses of treatment comparisons and synthesizing proportions [37]. The raw proportion (no conversion involved) for each outcome from all meta-analysis cohorts with at least one of the three outcomes (asphyxia, trauma, and hypothermia) was used. A random-effects GLMM was employed to generate pooled estimates of effect, because (following the Cochrane Handbook [38] and [37]) we regarded the analysed studies as being sampled from a universe of possible studies, as they had multiple populations, different methods to determine mechanism of death, and geographical variation. To stabilize the variance and normalize the distribution of these proportions, logit transformation was used to treat the raw proportion for entering the model [39]. The pooled proportion was expressed as proportion estimate with accompanying 95% confidence intervals (CIs). Clopper-Pearson method was used to compute confidence interval for individual studies [40]. Overall pooled proportion estimates and the contribution of individual cohort for each cause of avalanche death were visualized as forest plots (see [41] for more on this plotting method). Key findings from all subgroup meta-syntheses were systematically tabulated into a comprehensive table, including detailed information on subgroup proportion estimates, confidence intervals, and heterogeneity metrics (see section "assessment of heterogeneity").

### Sensitivity analysis

To assess the robustness and stability of the syntheses, leave-one-out sensitivity analyses were performed for each cause. We systematically excluded one study at a time and estimated PCAD for the remainder, to evaluate the influence of each individual study on the overall pooled effect size estimate and its precision.

Sensitivity analyses were conducted to deal with: *a*. studies with overlapping data (defined in *Data Extraction* section), *b*. studies with varying risk-of-bias, and *c*. studies potentially influenced by commercial interests. We did this by: *a*. keeping one single study with largest sample size for each overlapped group of studies, *b*. removing studies with high risk-of-bias rating, and *c*. removing studies potentially influenced by commercial connections, respectively.

## Assessment of heterogeneity

Heterogeneity in meta-analysis refers to the variation in study outcomes between studies [38]. A pooled estimate with high heterogeneity indicates the estimate should be interpreted with caution. In the present study, we followed the method proposed by Borenstein *et al.* [42] and evaluated heterogeneity by combining Prediction interval (PI) (Prediction interval provides a range within which we can expect the effect size of a new study to fall with a certain level of confidence) and $I^2$ ($I^2$ quantifies the percentage of variation in the pooled proportion that was due to observed variation between studies rather than sampling error). Specifically, a high-to-moderate percentage of variation (indicated by $I^2$) in a wide dispersion of true proportions (indicated by PI) indicates high heterogeneity; a low percentage of variation in a moderately-to-very wide dispersion of true proportions is at least moderate heterogeneity; a low-to-high percentage of variation in a narrow dispersion of true proportion is low heterogeneity; a negligible percentage of variation in any dispersion is low heterogeneity. For detailed explanation, see [42]. We evaluated the magnitude of $I^2$ by the common threshold of 50% ($I^2$<50% indicates low heterogeneity, $I^2 \geq$50% indicates moderate-to-substantial heterogeneity [43]). The determination of narrow or wide PI is case-by-case. In our case, a cause of death with higher proportion has more room to vary and is reasonably associated with a wider PI. Considering the commonly-reported proportions for the causes, we define a PI as wide when the length is wider than 0.2 for trauma, 0.4 for asphyxia and 0.1 for hypothermia; as moderate when the length is between 0.1 to 0.2 for trauma, 0.2 to 0.4 for asphyxia, and 0.05 to 0.1 for hypothermia; length even shorter than the corresponding lower ends were considered narrow. (We deviate from our protocol: in the registered protocol we planned to assess heterogeneity with $I^2$ alone. The current practice lowers the risk in observing false meaningful subgroups.)

We reported $Tau^2$ ($Tau^2$ is an absolute indicator for between-study heterogeneity; PI inherently accounts for $Tau^2$: the length of the PI is approximately 4 $Tau$ if within-study variances are small [44], hence we considered it as an auxiliary reference) and within-subgroup Cochran's Q statistics (Within-subgroup Cochran's Q statistic evaluates the heterogeneity within each predefined subgroup of studies by whether variation in effect sizes is greater than would be expected by chance.) as auxiliary references for heterogeneity. We refer to these auxiliary references only in cases when PI and $I^2$ were inconclusive for assessing heterogeneity.

## Subgroup analysis

Subgroup analysis was performed if there was unexplained high heterogeneity for the initial pooled proportion estimates, and if sufficient data was available. The primary focus was on data *time period*, *regions*, *sample sizes*, *data representativeness*, and *forensic diagnostic procedures*.

*Time period* is defined as the range of years of data described in a study. We stratified the time period before and after the year of 2000, considering there was crucial medical and technical progress during mid- to late-1990s. It was not until mid-1990s that avalanche medicine became the subject of systematic scientific interest [45]. Until late 1990s, there was a lack of helpful recommendations or treatment algorithms in this regard, and in individual cases there was great uncertainty among mountain rescue doctors when making decisions [45]. Additionally, the first digital avalanche transceiver was made in 1997 [46]. There is also a legitimate argument about the methodological differences between several very early studies in the domain before 1970s and studies published afterwards [14]. Besides, some studies used data covering an inseparable period across the year of 2000. We hence categorised the time period as '*before 1970*', '*between 1970-2000*', '*after 2000*', and '*across 2000*'.

***Regions*** are the countries where the accidents happened. European countries were analyzed alone and jointly in subgroup analysis. Other countries were analyzed alone.

***Sample size*** categories include <30, 31 – 75 and >75, defined as the number of victims included in a study or a cohort for analysis (a study can report multiple cohorts, usually for the purpose of comparison). The cutoff was defined according to a popular rule of thumb for sample size (n=30) [47] and minimum sample size for detecting hypothermia (n = 75), which has the lowest proportion among the causes.

This minimum sample size was calculated using the following formula: assuming the proportion of hypothermia is 2%, and the desired margin of error is 1%. The sample size *n* required to detect a prevalence of 1%:

$$n = \frac{Z^2 \times p \times (1 - p)}{E^2} \tag{1}$$

where:

- *Z* is the Z-value (1.96 for 95% confidence)
- *p* is the expected prevalence (0.01 for 1%)
- *E* is the desired margin of error

***Data representativeness*** is the size of the population from which the samples were obtained. Populations can be: *a.* Hospital/institution-based, *b.* local, *c.* nationwide, or *d.* multi-nationwide. This was determined by the region coverage of each data source.

The ***forensic practices*** or evaluation method leading to primary cause diagnosis varied. Diagnoses in a sample can be: *a.* performed in a strategic procedure (first macroscopic (external) examination and then autopsy for cases if necessary) or performed internal autopsy for all cases; *b.* macroscopic (external) examination alone; *c.* mixed between *a.* and *b.* (e.g. due to family members' refusal to permit autopsy); and *d.* decisions of the authors (of the included studies) based on accident/medical record. We hence designated this autopsy diagnostic subgroup including strategic/full autopsy (sequential or all internal autopsy), external alone, mixed and authors. (This is an inconsistency with the pre-registered protocol: we did not foresee differences in autopsy practices and this subgroup is not registered.)

To test whether the effect sizes differ significantly between subgroups, separate meta-analyses for each subgroup were performed and between-subgroup heterogeneity was tested using Cochran's Q statistic. We reported the resulting two-sided *p*-value [48] with its statistical power [49]. *p*<0.05 suggests the subgroup is associated with the source of heterogeneity. However, recognizing the method's sensitivity to sample size and our limited number of included studies, we did not consider the result reliable unless its statistical power is greater than 80%. In such a case, we took each subgroup's heterogeneity decreasing from the heterogeneity of overall proportion estimates as evidence for between-subgroup difference.

## Qualitative systematic review

We complied a qualitative systematic review based on all the included studies (meta-analysis and SR-only studies). Specific content for the review were determined in a post-hoc manner, after full-text investigation of the included articles. The review is presented in *Study characteristics and systematic review* section.

### External validity

Two-sample tests for equality of proportions were made to compare the pooled estimates with results of external studies (selected from the biased sample subcategory of SR-only studies). We hypothesized that we will obtain sensible and intuitive results in these comparisons. They would provide preliminary evidence on the divergent validity of our summary proportion and the risk/protective factors of types of avalanche deaths. A *p*-value <0.05 was considered statistically significant.

### Other statistical practices

In the present study, we did not try to produce summary estimates for the proportion of sports involved in avalanche deaths, considering some of the sports such as snowmobiling and heli-skiing undergo varied regulations across countries, or within regions of the same country, creating substantial heterogeneity. [50] Instead, descriptive statistics were performed to show frequency/proportion of different type of sports involving in the fatal avalanche accidents, using data from all included studies, if available.

### Result presentation

Study selection was presented in a flowchart. The results of overall proportion estimates were presented as forest plots. Risk of bias for meta-analysis studies was visualized as risk-of-bias plot.

Description for all included studies, key results of subgroup proportion estimates, key results of external validity tests, and proportions of activities preceding the accidents for each study, are tabulated as tables.

### Tools

We used Zotero [51] to manage retrieved articles. The meta-analysis was conducted using the $\mathbb{R}$ platform for statistical computing [52]. Specifically, we used the *metafor* package [53] for performing all GLMM-based meta-synthesis and generating forest plots. We used *ggplot2* [54] to generate the risk-of-bias plot. We used base R functions to perform two-sample tests for equality of proportions and sample size calculations (for checking the proper sample size needed for hypothermia).

## Results

### Study selection

The flow of study selection is charted in Fig 1. Initial searches returned 1,451 records plus three registries. After removing duplicate records (n=701), 750 titles and abstracts plus three registries were manually skimmed, resulting in 139 potentially relevant articles. The registries were excluded since none contained decisive diagnoses of causes for avalanche death. The article full-texts were retrieved, resulting in 26 eligible studies (18 for meta-analysis and systematic review, and eight additional studies for SR-only). Among the included studies, one reported causes of avalanche death in Japanese skiing resorts without giving specific numbers [55], but the missing cause of death information was obtained from the corresponding authors. Another study [56] had reported inconsistent numbers for different causes of avalanche death and total number of all fatalities, yet we could not contact the author, and the paper was included for SR-only. For more details of contacting authors, see S3 Table. Screening details for all included and excluded studies are available on S4 Table.

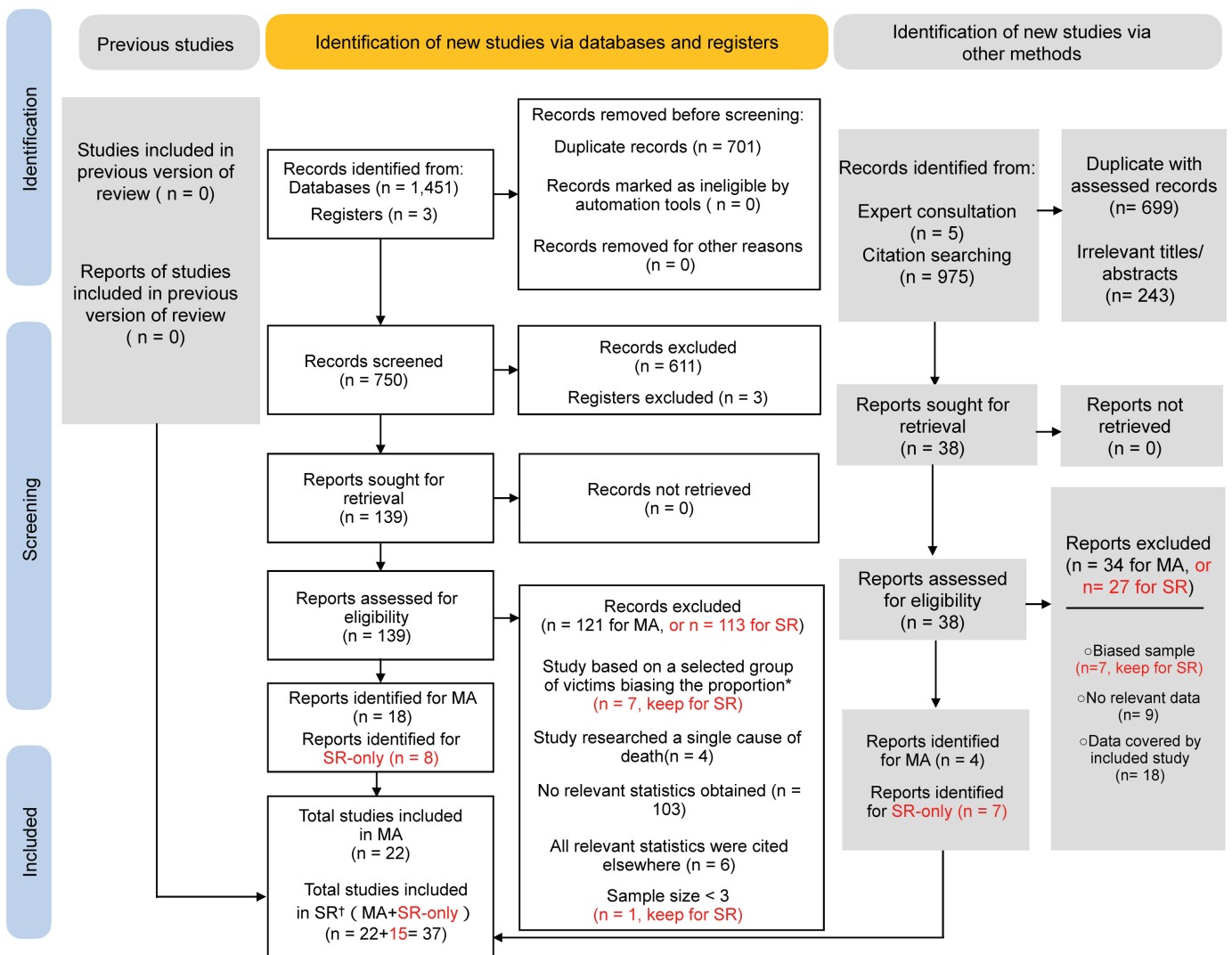

**Fig 1. Flowchart of the study selection process.** Note: MA: meta-analysis; SR: systematic review; SR-only: systematic review only (highlighted in red).

The initial database search was followed up in two ways. First, we checked reference lists of the 26 studies (n=975). Second, we consulted domain experts, who provided us with five potential studies/registries. These together produced 20 potential studies for which we assessed the full texts. From them we identified three studies and one register for meta-analysis (n = 4) and seven studies for SR-only.

Some studies appeared to meet the included criteria but were still excluded, mostly because their numbers or statistics were cited from other studies, and we eventually found and included the original studies [14,57,58].

We made several critical decisions in the process. One study [59] reported presumptive diagnoses of 596 avalanche victims by their NACA severity score. We decided to exclude it by two considerations: *a.* although NACA score of 7 means on-site death, patients with NACA severity scores of 5 or 6 also risk failure of surviving in emergency department after hospitalization (the study did not follow up to in-hospital stage), biasing the results towards

on-site death; *b.* according to the authors, the diagnoses were presumptive diagnosis of the victims' on-site conditions (even for the dead) and it is intrinsically different from diagnosis of primary cause of death, especially considering the system has 'hypothermia' as default [59]. We thus excluded the study from meta-analysis for reason *a.*, and from systematic review for reason *b.*

One study [60] has exceptionally high proportion of cases without cause of death diagnosis (34%). The authors of the study suspected that the non-investigated cases in the data may include fewer cases with trauma-related deaths [60]. We considered this can substantially bias the results and hence included the study in SR-only. Similarly, most of the fatality data included in another study [61] were collected from media reports from various sources, risking high ratio of missing cases and inconsistent causes-of-death determination methods. We included it in SR-only. Both studies are grey literature.

On the other hand, some studies [62,63] focused on PCAD for victims doing specific types of sports, such as skiing. Since current evidence does not reveal different percentage of PCAD with these activities, we hence decided to include them, but gave special attention to their influence on pooled estimates and heterogeneity during LOO sensitivity analysis. In contrast, since existing evidence has established the association between burial duration (or extrication time) and depth with increasing risk of death by asphyxia [64], studies selecting fatalities under the influence of these factors were included as SR-only, rather than MA. This encompassed one study aiming at fatalities with long burial [33], two studies [30,32] focusing on full burial fatalities, one study [55] looking at avalanche fatalities in ski resorts (shorter extrication time), and one study [65] analyzing fatalities due to summer avalanches (apparently shallower snow).

As such, 37 studies were included in the systematic review and 22 of these in the meta-analysis. The latter 22 studies included two registries [66,67], 19 published studies [11,18, 19,22,26,27,34,62,63,68–76] and one unpublished study [77] for meta-analysis and systematic review. The 15 in SR-only were 14 published studies [12,21,30,32,33,55,56,60,65,78–82] and one unpublished study [61]. Twenty-seven studies were in English language only and five were in German [12,19,21,30,69], three in French [56,68,70], one in Norwegian [21], and one having both English and Polish versions [81].

The registries were for Canada [66] and New Zealand [67]. The Canadian register includes accidents from 1996 to 2007. Since a section of this data (1984 to 2005) has already been covered by another cohort [26] which used a more reliable method to determine causes of death, our study only obtained register data from 2006 and 2007, to prevent duplication of cases. The New Zealand register did not have causes of death for all cases, but we included it because the cases having causes of death diagnosis were random rather than arbitrary [67].

## Data extraction

We contacted the corresponding authors of one study for missing causes of death [71]. We failed to obtain any responses. For more details, see S3 Table.

Data for analysis are available on S5 Table. A summary of extracted data is presented in Table 1.

## Study characteristics and systematic review

**Selected cohorts.** One of the included studies [22] reported PCAD for three separate cohorts (USA, Canada and Europe), they were treated as three individual entries for meta-analysis, resulting in 24 cohorts with an overall sample size of 1,550 for the meta-analysis. The additional 15 studies for systematic review provide 897 more samples.

Table 1. Study characteristics and data for analysis.

| Publication* | Sample§ Years | n sample (n males) | Source | Feature | Location | Design|| | Burial† % of type (Complete) | Time (min) | Depth (m) | Forensic/evaluation‡ Diagnosis (n) | method |
|---|---|---|---|---|---|---|---|---|---|---|---|
| Studies for meta-analysis and systematic reviews | | | | | | | | | | | |
| Alnoncourt, 2017 [68] | 2010–2017 | 25(24) | Records of victims autopsied at the medico–legal institute of Grenoble University of hospital | All autopsied | Grenoble, France | R | 92% | M [R]: 954[10–17280] | 1.467[0.2, 3] | T(1), A+T (6), A (18), H (0) | F/S |
| Bilek and Würtl, 2011 [69] | 2005–2011 | 143(n.r.) | Police report | All | Austria (specific location unclear) | R | n.r. | n.r. | n.r. | T (46), S (83), H (2), U (12) | mixed |
| Boyd, 2009 [26] | 1984–2005 | 204(179) | British Columbia Coroners Service and the Chief Medical Examiner of Alberta+the database of the Canadian Avalanche Centre for fatal avalanche details. | All | British Columbia and Alberta | R | 167/204 (82%) | M R: Trauma: 25[10–56] Asphyxia: 45[20–231] | M R: Trauma: 0.9[0.3–1.2] Asphyxia: 1.5[1–2] | T (48), A (154), H (2) | Mixed |
| Christensen, 1999 [62] | 1977–1997 | 12(n.r.) | Case files of the Pierce County Medical Examiner's Office | Mountaineerer | Mount Rainier, USA | R | n.r. | x̄:109 M: 97.5 | n.r. | T (2), A (10), H (0) | F/S |
| Degawa, 2023 [77] | 1991–2020 | 196(n.r.) | Japan Avalanche Network Database | All | Japan | R | 79% | n.r. | R: [1.1, 1.5] | T (29), S (126), H (8), U(33) | n.r |
| Eliakis, 1974 [70] | n.r | 24(n.r.) | n.r | All | Switzerland | n.r | n.r. | n.r. | n.r | T (1), A (21), H (2) | n.r |
| Fredriksen 2013 [83] | 1996–2012 | 48(37), | Reports from the rescue teams and the medical records from institutions | All | Northern Norway and Spitzbergen | R | 28(58%) | n.r. | n.r. | T (8), A (22), H (2), D (5), CA(2), U(9) | Mixed |
| Gross, 2021 [71] | 2001–2019 | 32(n.r.) | Records from the Institute of Forensic Medicine of the University of Berne; | All | Canton of Berne, Swizerland | R | n.r. | n.r. | n.r. | A (22), U (10) | Mixed |

(Continued)

**Table 1.** (Continued)

| Study | Period | n | Source | Inclusion | Location | Type | % | Age | Sex | Causes | Setting |
|---|---|---|---|---|---|---|---|---|---|---|---|
| Grossman, 1989 (1) [22] | 1982–1987 | 12(n.r.) | Records from US forest service. | All | Utah, Salt Lake City, USA | R | 11/12 (92) | x̄:74 (n = 7) | x̄: 5.4 (n=7) | T (2), A (10), H (0) | Mixed |
| Grossman, 1989 (2) [22] | 1975–1985 | 378(n.r.) | Records of International commission on Alpine rescue | All | France, Switzerland, Austria, and Italy | R | Both | n.r. | n.r. | T (97), A (273), H (8) | n.r. |
| Grossman, 1989 (3) [22] | 1979–1985 | 45(n.r.) | Database of Canadian avalanche association | All | Canada | R | Both | n.r. | n.r. | T (12), A (28), H (0) | n.r. |
| Hohlrieder, 2007 [27] | 1996–2005 | 36(n.r.) | Records from Hospital of Innsbruck was used. | All | Innsbruck, Austria | R | Both | n.r. | n.r. | T (2), A (33), H (1) | F/S |
| Irwin, 2002 [67] | 1981–1997 | 19(n.r.) | Aotearoa, New Zealand | All | Aotearoa, New Zealand | Re | 94% | M 45 | x̄ 1.58 M: 1.5 | A (9), T+A (6), T (1), T+H(1), T+A+H(1), H (1) | n.r |
| Jamieson, 2007 [66] | 2006–2007 | 10(n.r.) | Databases of the Canadian Avalanche Centre | All | Canada | Re | Both | n.r. | n.r. | T(2), A(8), H (0) | n.r. |
| Johnson, 2001 [34] | 1992–1999 | 28(n.r.) | Records of the state medical examiner | All | Utha, USA | R | n.r. | n.r. | n.r. | T(6), A(22) | n.r. |
| Lugger and Unterdorfer, 1972 [19] | 1964–1970 | 20(14) | Affiliated hospital of Institute for Forensic Medicine at the University of Innsbruck | All | Innsbruck, Austria | R | n.r. | n.r. | n.r. | T (1), A (19) | F/S |
| Martinez, 2022 [72] | 1971–2020 | 48(38) | Hospital Arnau de Vilanova | All autopsied | Spain | R | 57.1% | n.r. | n.r. | A (24), T(14), H (4), U(6) | Mixed |
| McIntosh, 2007 [18] | 1989–2006 | 56(53) | Utah Avalanche Center and Medical Examiner records | All | Salt lake city, Utah, USA | R | n.r. | n.r. | n.r. | T (3), A (48), H (0), T + A (5) | Mixed |
| McIntosh, 2019 [73] | 2007–2018 | 32(30) | Utah Avalanche Center and Medical Examiner records | All | Salt lake city, Utah, USA | R | n.r. | n.r. | n.r. | T (6), A (23), H (0), T + A (3) | Mixed |

(Continued)

**Table 1.** (Continued)

| Study | Period | n (deaths) | Data source | Population / Limitation | Location | Design | Completeness | Maximum | Central value | Causes | Classification |
|---|---|---|---|---|---|---|---|---|---|---|---|
| Moroder, 2015 [74] | 2008–2013 | 13(12) | Accident reports of the Tyrolean Avalanche Forecast Centre. | All | Tyrol, Austria | R | 3/10(30%) | Maximum: 35 | n.r. | T (3), A (7), U (3) | Mixed |
| Oshiro, 2022 [75] | 2011–2015 | 26(n.r.) | Official medical record | All | Japan | R | n.r. | M: 150 | n.r. | T (4), A (22), H (0) | External |
| Sheets, 2018 [76] | 1994–2015 | 110(n.r.) | Database maintained by the Colorado Avalanche Information Center | All | Colorado, United States | R | n.r. | n.r. | n.r. | T (32), A (74), H (4) | Mixed |
| Stalsberg, 1989 [11] | 1986 &1987 | 18(n.r.) | Hospital medical record | All | Norway | CS | Both | x̄:1095.6 M: 330 | x̄:1.7 M:1.75 | T (2), A (15), D (1), H (0) | Mixed |
| Tough, 1993 [63] | 1980–1991 | 15(14) | Official medical record | Skier | Alberta, Canada | CS | n.r. | n.r. | n.r. | A + T (15), H (0) | F/S |
| **Studies for systematic reviews only** | | | | | | | | | | | |
| Blancher, 2017 [78] | 2017 | 6(n.r.) | Rescue case record | Single accident | Valfrejus, Savoie, France | R | Both | n.r. | n.r. | A (5), U (1) | F/S |
| Eidenbenz, 2021 [33] | 1997–2018 | 67(n.r.) | Swiss Institute for Snow and Avalanche Research (SLF) registry in Davos | Long–burial | Switzerland | R | Complete | R: >60min & <24h | M (IQR): 1(0.65 to 170) | T (9), A (41), H (9), U (8) | Authors |
| Geisenberger, 2015 [21] | 2015 | 2(1) | Hospital medical record | Single accident | Southern Black Forest, Germany | R | Complete | 120 min for both | 1m for 1 victim, others unknown | A+H (2) | F/S |
| Grosse, 2007† [79] | n.r, 2 years | 2(n.r.) | Records from trauma center of switzerland | Small sample | Switzerland | P | Complete | x̄: 57.5 | median:1.5 | A+H (2) | Mixed |
| Haegeli, 2011 [32] | 1980–2005 | 143(n.r.) | Databases of the Canadian Avalanche Centre | Complete burial | Canada | R | Complete | M:25 | n.r. | T (27), A (116), H (0) | Mixed |
| Hatwal, 2021 [80] | 2019 | 12(n.r.) | Hospital where the forensic diagnoses were made | Single accident | India | R | n.r. | 20–50 days | n.r. | T (12), A (0), H (0) | F/S |
| Kobek, 2016 [81] | 2003 | 6(4) | Hospital where the forensic diagnoses were made | Single accident | Poland | C | n.r. | 106–141 days | n.r. | T (0), A (6), H (0) | F/S |
| Lapras, 1980 [56] | 1975–1976 | 41(n.r.) | n.r | Inconsistent number reporting | French | R | 40/41(98%) | M R: 60–180 | M R: 0.50–1 m | T (11), A (16), H (2), D (1), U(1) | n.r |

*(Continued)*

**Table 1.** (Continued)

| Study | Period | Sample | | Inclusion notes | Country | Design | Type | Time | Depth | Diagnosis | Method |
|---|---|---|---|---|---|---|---|---|---|---|---|
| Locher, 1996 [12] | 1980–1987 | 16(n.r.) | n.r | All with signs of hypothermia | Switzerland | R | n.r. | x̄: 78.2 (n = 15) | n.r. | T (0), A (8), H (8) | Mixed |
| Mair, 2013 [82] | 2008–2011 | 15(n.r.) | HEMS rescue missions conducted for avalanche accidents | on–site deaths excluded | Tyrol, Austria | R | n.r. | n.r. | n.r. | T (5), H (3), U (7) | n.r. |
| Markwalder, 1970 [30] | Before 1969 | 43(n.r.) | n.r | Full burial | Swiss | R | Full | M: 135 | M: 1.5 | T (5), A (21), H (1), T + A (2), PS (9), U (5) | F/S |
| Pasquier, 2017 [65] | 1984–2014 | 35(n.r.) | Swiss Institute for Snow and Avalanche Research registry in Davos | Summer avalanche | Switzerland | R | Both | x̄: 1156 (n = 10) | x̄: 7 | T(33), D (1), A (1) | Authors |
| Tanaka, 2024 [55] | 2011– 2023 | 5 (n.r.) | Ski Resort Injury Report data | Ski resort | Japan | R | Both | n.r. | n.r. | S (1), Possibly S (4) | n.r. |
| Techel and Zweifel, 2013 [60] | 1936 –2012 | 417(n.r.) | Avalanches reported to SLF stored as avalanche database | High ratio of non-investigated cases | Switzerland | R | Both | M: 64min | M: 0.1 | T(115), A(152), H (8) | Mixed |
| Zachau, 2020 [61] | 2000–2018 | 23(n.r.) | Media and other sources | Media reported accidents | Newzealand | R | n.r. | n.r. | n.r. | T (9), A +T(5), A (6), H (1), U (2) | n.r. |

(*Continued*)

* All the data was extracted by RG and verified by LA. Data of studies [66,67] was extracted on April 28th; Data of studies [69,77] was extracted on May 25th, 2024. Data of studies [12,21,28,30,32,33,55,56,61,65,78–82] was extracted on May 26th, 2024. Data of studies [11,18,19,22,26,34,62,63,68,70–76,83] was extracted on May 15th, 2024; Data of studies [69,77] was extracted on May 25th, 2024. Data of studies [12,21,28,30,32,33,55,56,61,65,78–82] was extracted on May 26th, 2024.

§Under Sample columns: column "n (n of man)" is sample size and sample size for male; n.r is not reported.

|| Under Design column: n.r: "not reported"; R: Retrospective; P: Prospective; C: Case report; CS: Case series.

†Column "% of type (complete)" is to record the percentage of complete burial; In the cells, "Complete" means all fatalities had complete burial; "Both" means some fatalities had complete while others had partial burial, and proportion is not reported in the study; When a percentage is reported alone, it is the percentage of complete burial; For columns "Time (min)" and Depth (m), the units are minutes and meters, respectively, unless otherwise specified. M is median; x̄ is mean; R is range; MR is the median range when the study reported numbers of fatalities in several ranges;

‡"Diagnosis" is cause of death diagnosis. T is trauma; A is asphyxia; H is hypothermia; D is drowning; U is unclear. "Method" is the forensic diagnosing method. "Mixed" is when some cases underwent external examination alone, while others underwent either autopsy or both, due to NON-medical considerations such as refusal of fatalities' family; "F/S" means "Full or strategic". It is when either all cases underwent full autopsy (internal or internal+external autopsy) or they received either internal or external autopsy, according to medical considerations. "External" is external examination was performed. "Author" is when the authors determined cause of death case by case according the accident records or reports which do not include an existing cause of death diagnosis. "Martinez, 2022" selectively chose all the cases with autopsy from hospital records for reporting. The procedure is distinct from full/strategic procedures used in other studies, where full cases over a period of time are used. We hence categorized the study into mixed-autopsy subgroup.

**Causes of death.** Among the 24 meta-analysis cohorts, proportions of trauma were extracted from 23 cohorts; proportions of asphyxia were extracted from 22; and proportions of hypothermia were extracted from 22 cohorts.

Though asphyxia, trauma and hypothermia are commonly used causes of death categorization (referred to as the Common Three), variations were observed in included cohorts. For one cohort [34], the medical examiners that made the forensic diagnoses differentiated all fatalities into either trauma or asphyxia (without considering hypothermia). One cohort [22] was not clear about whether hypothermia was considered as a cause. Two cohorts [11,83] included drowning as cause of death. One cohort [83] included cardiac arrest. (Drowning was combined into asphyxia in the meta-synthesis; cardiac arrest was removed from analysis) Two studies [69,77] employed the cause of "suffocation" in place of asphyxia. (Though suffocation and asphyxia can be used interchangeably in some contexts, distinction should be noticed. 'Suffocation' might focus on a specific subset of asphyxial deaths where mechanical obstruction plays a primary role [17], such as snow absorption. Other subsets like death due to elevated carbon dioxide levels with air-pocket underneath snow can be excluded. To ensure consistency and accuracy, we decided to exclude these two studies from pooled/subgroup proportion estimates for asphyxia.) Cases with causes of death being categorised as 'unclear' were observed in two cohorts [74,83]. Notably, a 1974 study is earliest among all included studies in using the Common Three as causes of death categorization [70].

Forensic diagnoses in the included studies sometimes were a combination of multiple causes: asphyxia and trauma [68], trauma and hypothermia [67], or traumatic asphyxia and hypothermia [67].

Subcategory of avalanche death due to trauma was reported in four cohorts [18,26,34,79]. One cohort found chest trauma was the most common injury, representing 46% (11/24) of cases with single-system trauma, followed by head injury, representing 42% (10/24) of the cases with single-system trauma [26]. In one cohort, 61% (17/28) of victims had some degree of CHI based on postmortem examination; 21% of victims (6/28) had severe head injuries and they were noted by the medical examiner as either contributing to death or being the primary cause of death [34]. In one cohort, the authors reported that all fatalities in their data attributed solely to blunt trauma (n=3) had evidence of head injury [18].

In two cohorts [62,79] information on trauma subcategory was only available for a mixed population where avalanche fatalities were included. One cohort reported 28 types of injuries observed in 14 avalanche victims including both survivors (n=12) and fatalities (n=2) [79], among which chest injuries accounted for 64%, followed by head injuries for 28%. One cohort [62] reported the subcategory of 20 trauma fatalities. Among them, multiple injuries including head, chest, and abdomen accounted for 40% (n=12), followed by head injuries for 23% (n=7), and chest injuries for 1% (n=1). We highlight the importance of clarifying criteria for forensic diagnosis, since the presence of chest injuries can be a sign of death by either compression asphyxia or traumatic injuries and different criteria can lead to different diagnosis categories. However, this was insufficiently done by some studies.

For SR-only studies, five used forensic diagnostic frameworks including at least the Common Three [28,30,32,33,56,61]. Notably, in one early study published in 1970 [30], psychogenic shock (n = 9) was used as a cause of avalanche death, except for the Common Three. This cause was never used again in the later studies. Many studies only had one or two causes of deaths reported among the Common Three, most possibly due to small sample sizes [21,55, 78,79] and/or the nature of a single avalanche [21,80,81]. Biased samples in SR-only studies can be a limiting factor to observing all Common Three. One study [12] collected victims with signs of hypothermia to establish differential diagnosis criteria for circulatory failure. The trauma cause of death was hence not detected. One study used a sample of fatalities who

died on-site and no asphyxia death was reported. One study specifically looked at fatalities of summer avalanches, and no hypothermic death was reported. One Canadian study used a large sample of fatalities undergoing complete burial (n = 143) but no hypothermic death was detected. In contrast, the 1970 Swiss study also examined complete burial cases alone, and found one out of 43 died from hypothermia.

**Demographics of avalanche fatalities.**   For meta-analysis studies where gender ratio were provided [18,19,26,63,68,72–74,83], males accounted for 70% [68] to 96% [19] of the fatalities. In the studies for SR-only, males made up 50% [21] to 67% [81] of fatalities. One study [63] pointed out the preponderance of male victims can be the result that backcountry winter sport is more popular among males. Another study [73] argued this may be partly due to the demographics of backcountry users and increased risk-taking behavior in male participants. However, none of the included studies investigated the role of gender in shaping PCAD.

A Canadian study published in 1993 reported demographics of 19 fatalities and found an average age of 36 years (42% in their 20s, 36% in their 30s or 40s) is older than that typically associated with risk-taking behavior [63], and the authors believed the wide range of age suggests that factors other than age play a role in backcountry fatalities [63]. The age pattern was echoed by a 2007 Swiss study [79], which reported a mean age of 37.4 and a wide range of 17–59 for the studied victims, and by another Canadian study [26] using 21 years of data, which found the median age of the studied fatalities was 33 (interquartile range 26–43) years. Similarly, an American study [18] published in 2007 reported a mean age of 31 (SD 10) with a range of 7 to 59. In contrast, evidence from Japan showed a pattern towards older age. A study [75] analyzed Japanese avalanche fatalities between 2011 and 2015, finding a higher mean age of 48 with an age range of 39 to 58.3 years. Compared to studies above, this range had a similar upper limit but a much higher lower limit. A more recent study collected Japanese fatalities covering a wider time period of 1991 to 2020, and reported a mean age of 42 (median: 41) with 50% of the fatalities in their 30s and 40s. Conversely, a Polish study [81] described six fatalities from one avalanche, with five victims aged 17 or 18, but they were from one secondary school expedition and may not be extropolative. A noteworthy study by Sheets et al. examined the relationship between age and trauma-related deaths, finding no significant differences across age groups [76]. While other direct evidence on age's effect on PCAD is not available, we reused data from Tough et al. [63] for a Bayesian estimation of two groups (This is a Bayesian equivalent of the t-test proposed by Kruschke [84], adopted due to the small sample size (n = 19)). No strong evidence was revealed for a significant difference in age between deaths by trauma and asphyxia (95% Highest Density Interval for the difference in means: −32.7 to 30.8).

**Study design and data source.**   For the meta-analysis cohorts, two [11,63] were case-series; two [66,67] were extracted from registry; the other 20 were retrospective, using either avalanche accident databases [18,22,73,74,76,77], case files (preserved by university [71], hospital [19,27,68,72], rescue center [22,83], police system [69] or medical examiners [34, 62,75]), or combined [26]. One study [70] did not clarify their study design. The database used as data source included Database of Canadian Avalanche Association [22], Database of Utah Avalanche Center [18,73], Database of Tyrolean Avalanche Forecast Centre [74], and a database maintained by the Colorado Avalanche Information Center [76].

For SR-only cohorts, one [79] claimed they prospectively performed autopsy in a local trauma center (the study was not registered before-hand); one [61] mainly used accidents reported by social media; three [12,21,56] did not clarify their study design; others were retrospective, using either avalanche accident databases [32,33,55,65] or case files (preserved by rescuer [78,82] and hospital [80,81]). The databases used include the Swiss Institute for

Snow and Avalanche Research (SLF) register in Davos [33,65], Databases of the Canadian Avalanche Centre [32], and the database from the Japanese Association for Skiing Safety [55].

**Ski/avalanche-forecasting expertise/experience and mitigation equipment of fatalities.** Information about fatalities' expertise in skiing was sparse in the included studies. One study [22] and one registry [22] reported fatalities involving expert back-country skiers or avalanche controllers. One study [67] investigated the role of skiing experience in carrying complete equipment.

Skiing and avalanche forecasting expertise may lower the risk of running into and/or getting caught by a fatal avalanche (e.g. through better planning [71], better risk assessment and decision-making [71], snow-cutting technique [80]), and increase the chance to survive it (e.g. through avoiding "anchor effect" of skis [34], self-rescue skills such as 'swimming' to the surface of the slide [27,34], or creating a breathing space around the chest and face [26]). Notably, the preventability of the causes of death can vary with type and level of expertise. However, none of these studies sought evidence on the association of Ski/avalanche-forecasting expertise/experience and PCAD.

Mitigation equipment used by fatalities was widely discussed [18,22,26,27,34,73,79]. Different types of avalanche mitigation gear are specifically effective in preventing different causes of death. Some gear is designed to prevent trauma, such as helmet [18,26,34,69], while others are better suited for preventing asphyxia such as avalanche airbags [27,73,76,79]. However, the focus of the included studies was placed on their role in reducing overall fatalities. None of these studies provided evidence on the association of mitigation equipment and PCAD.

**Weather and snow.** Recreational avalanche accidents most often occur in dry-snow conditions (90%) [28], 62% of avalanche resulting in fatalities are slab avalanches [77] and fluctuations of avalanche accidents over years can be explained by meteorological variations such as snow and weather conditions [71]. These variations also affect the outcome of mountain rescue [75].

Weather and snow's roles in PCAD were investigated in several studies. In one study, an established concept of snow climate (including maritime, transitional and continental climates) [85] was used as grouping factor to examine its effect on PCAD, and the results revealed, while overall mortality did not differ significantly between the climates, the proportion of trauma-related deaths was significantly higher in the transitional snow climate than in the other two snow climates [32]. Significant differences in avalanche survival curves by snow climate were also observed [32]. Pasquier et al. investigated the epidemiology of summer avalanche accidents that occurred in Switzerland between 1984 and 2014 and found poly-trauma became the leading cause of death in summer avalanche [65]. The authors ascribed it to very steep and exposed terrain involved in summer avalanche [65], rather than weather and snow factors such as shallower burial. By analyzing the cause of death of 24 avalanche fatalities, Eliakis [70] argued powder snow avalanches may attain very high, even supersonic speeds, and barotrauma may be caused in such cases by the wave of positive pressure travelling in front of the avalanche and the following wave of negative pressure, causing tympanic membrane rupture and pneumothorax. However, Stalsberg et al. [11] doubted the significance of the pressure wave as a cause of death.

Meanwhile, powder snow can be a protective factor to asphyxia. The idea was proposed by Grossman et al. [22], they argued power snow have a much greater oxygen content and it is more likely that a victim would be able to maneuver to the surface of the slide or dig out an air space. No empirical evidence was provided.

## Location and terrain

Regional differences in PCAD were observed but there is still dispute over the reasons. Proportion of avalanche death by trauma in United States was found to be 29% [76] or 28% [73], much higher than statistics from Japan such as 18% [77], or 15% [75], and also higher than statistics from European studies [11,19,27] and Canadian studies [63]. Some researchers attribute the pattern to terrain factors and argued avalanches in United States more often swept victims through or into trees, cliffs, rocks, and gullies due to its terrain forms [57,86]. However, a study that examined PCAD above and below the tree line revealed no increase in trauma at lower elevation [87]. Moreover, some studies reported low proportion of trauma for United States [18,22], and high proportion for Europe [69], contradicting the above terrain-trap argument. Sheets et al. suggested the lower rates of trauma in the European studies may reflect the varied research methods and lack of autopsy data more than differences in terrain or activity [76].

Two SR-only studies [55,82] investigated fatalities in specific locations. Tanaka et al. [55] examined three avalanche accidents involving five fatalities at Japanese ski resorts between 2011 and 2023, finding that all causes of death were either confirmed or possibly suffocation. The lack of autopsies and the small sample size may explain the pattern. Mair et al. [82] analyzed 15 on-site avalanche fatalities and found that none were due to asphyxia. The authors did not provide an interpretation for this pattern.

**Forensic diagnostic procedures.** The forensic practices leading to primary cause diagnosis were varied. See Table 1. Besides, one SR-only study [28] used data with causes of death diagnosed either on-site or sometimes by postmortem, and seven cohorts did not mention a forensic procedure. Notably, among the four all-autopsy studies, one [72] retrospectively selected the cases with autopsy from all cases they collected. This was a different practice from the other three, which prospectively autopsied the cases.

Insufficient reporting of autopsy findings in PCAD studies has long been recognized [11, 22]. Many authors of PCAD studies emphasized the importance of conducting autopsies or, at a minimum, thorough external forensic examinations to draw accurate postmortem conclusions [68,72,81,83]. One can doubt the validity of cause of death diagnosis in cases where an autopsy has not been carried out [21,83]. Sheets et al. [76] pointed out fatal internal traumatic hemorrhage may have only minimal external physical findings, and in this situation, asphyxia may be mistakenly assumed to be the cause of death.

Most studies in PCAD giving earliest evidence for autopsy findings of avalanche victims were published in German or French [19,30,56,70]. This can owe to the large numbers of people in Europe who live and work in mountainous regions than those in Canada or United States [22]. However, these studies suffered criticism of their methodological rigor, especially on sample size and sample representativeness [57]. Some English studies [11,22,63] published around 1990s had great improvement in methodological quality [57].

In the 21st century, PCAD research has branched into two directions: one focuses more on autopsy findings, while the other emphasizes the quality of methodological rigor. Studies elucidating the autopstic aspects of samples usually had small or biased samples. Several studies [21,78–81] gave detailed descriptions on forensic diagnostic procedures and findings but the samples used were either small or from one single avalanche accident, biasing the PCAD estimates. The importance of autopsy was further demonstrated in these studies. Kobek et al. [81] described how two avalanche fatalities with severe injuries were found

to have died from asphyxia through full autopsies, and the causes might otherwise be diagnosed as trauma. Geisenberger et al. [21] reported a case, whose cause of death was suspected to be trauma by the emergency doctor at the scene of the accident, was later confirmed as dying from asphyxia through autopsy. On the other direction, studies aimed to represent larger populations and used registries from national or regional databases, which usually have inconsistent and unclear diagnostic criteria for and recording practice of cause of avalanche death. Several PCAD studies [28,32,69,76,77,77] with large and representative samples were not able to give cause of death confirmed by autopsy for a considerable ratio of the samples and the forensic diagnostic criteria for the autopsied were underreported, limiting the credibility of the findings. Boyd et al. [26] analyzed 204 avalanche fatalities but an autopsy had only been performed on 117 of them. Degawa et al. [77] did not clarify if any autopsy was done to determine the causes of deaths in the sample. Meanwhile, some studies with smaller sample size also lacked sufficient practice and/or reporting of autopsy. In a 2001 study [34], autopsy was mentioned for some victims but the condition for others remained unclear. In a Japanese study, Oshiro et al. [75] clarified, for some cases in the study, autopsies were not performed because no crime and/or trouble was suspected. However, the specific number of such cases was not reported. As pointed out by Sheets et al. [76], the discrepancy between PCAD reported by these studies can be more possibly the result of insufficient autopsy practices than topological inconsistencies.

**Time periods and trending over time.**   Included studies covered avalanche accidents occurring in a wide time range from 1964 [19] to 2023 [55], and across 10 countries. Data for Poland and India were only available from SR-only studies. See Table 1. Notably, one study [22] analyzed a cohort mixing multiple European nations [22]. Advancements in avalanche rescue [28], treatment [25], safety equipment [46], and information on snow conditions and avalanche situations [1] have been ongoing, potentially impacting avalanche fatality rate. Investigating whether and how these improvements differently influence PCAD could provide valuable insights. However, none of the included studies have investigated the dynamics of PCAD over time.

**Fatality representativeness.**   Most cohorts included for meta-analysis represented avalanche fatalities in a period of time within a particular geographic region, either being hospital-based, local region of a country (town, city or state), national or multi-national. See Table 1.

In contrast, SR-only studies represented a selected/biased subset of fatalities (either because the study intentionally focused on a subset or the nature of the sample happened to do so). See Table 1. Notably, one cohort [56] had inconsistencies between the reported number of fatalities (n=41) and the numbers summed across all fatalities' death causes (n=31, the categories included 'unclear' cause of death), possibly indicating a typo and thus bias against the cause with mistakenly under-reported number.

Specifically, four studies [21,78,80,81] for SR-only were based on data from single avalanche accidents. The results indicated that victims in one accident tend to die from a same cause. Blancher et al. [78] reported five fatalities from asphyxia in one incident without providing an explanation. Geisenberger et a. [21] reported two fatalities from a combination of asphyxia and hypothermia, attributed partially to delayed rescue due to harsh weather. Similary, Kobek et al. [81] elaborated on the forensic diagnostic processes for six victims who died on site in an avalanche but whose body were found after months. The causes were diagnosed as asphyxia through full autopsy including histopathological studies. In contrast, Hatwal et al. [80] described an avalanche involving 12 mountaineers

all dying from trauma, likely due to falling from height while being carried away by the avalanche.

**Characteristics of burial-related factors for different causes of death.**   Burial type, time, and depth are known risk factors for different causes of death, with full and longer burial correlating with higher risk of asphyxia and hypothermia [14,64,88]. They can be a source of heterogeneity for meta-analysis and here we summarised relevant evidence from meta-analysis studies. See Table 1.

Though most meta-analysis cohorts reported that fatalities were either completely or partially buried, only seven clarified or gave information for computing the proportions, which ranged from 57.1% [72] to 94% [67]. Similarly, burial time for meta-analysis studies was under-reported and variable in terms of values (e.g. median 45 to 945 min) and summary statistics reported (mean, median, both or maximum) [67] (median 45 min). Burial depth was reported by six meta-analysis cohorts, varying from around 1.7 m (median) [11,26,67,68] to 5.4 m (mean) [22].

**Activities.**   Nine cohorts [12,30,56,60,62,74,75,77,82] did not report the type of activity; one cohort [21] consisted of only military personnel on an exercise; all other cohorts reported at least some avalanche fatalities who were doing winter sports, see Table 2.

Prior to the avalanche, individuals from included cohorts were active in snowboarding, multiple types of skiing (e.g. back-country skiing, ski touring, and heli-skiing), climbing (ice

**Table 2**. **Activities of the non-survivors preceding the accidents.**

| Publication* | Recreational (%) | | | Non-rec (%)† | Data‡ |
|---|---|---|---|---|---|
| | **With-ski** | **No-ski** | **Motorised** | | |
| **Studies for meta-analysis** | | | | | |
| Alnoncourt, 2017 | 100 | 0 | 0 | 0 | Unbiased (FRA) |
| Boyd, 2009 | 64 | 12 | 22 | 2 | Unbiased (CAN) |
| Degawa, 2023 | 45 | 0 | 3 | 0 | Unbiased (JPN) |
| Fredriksen 2013 | 51 | 49 | 0 | 0 | Unbiased (NOR) |
| Grossman,1989 (1) | 100 | 0 | 0 | 0 | Unbiased (USA) |
| Grossman,1989 (2) | 60 | 0 | 0 | 20 | Unbiased (EU) |
| Grossman,1989 (3) | 60 | 0 | 0 | 0 | Unbiased (CAN) |
| Irwin, 2002 | 25 | 56 | 0 | 19 | Unbiased (NZ) |
| Jamieson, 2007 | 57 | 29 | 0 | 14 | Unbiased (CAN) |
| Johnson, 2001 | 53 | 17 | 30 | 0 | Unbiased (USA) |
| Lugger, 1972 | 100 | 0 | 0 | 0 | Unbiased (AUT) |
| Martínezl, 2022 | 93 | 0 | 0 | 0 | Unbiased (ESP) |
| McIntosh, 2007 | 73 | 4 | 0 | 0 | Unbiased (USA) |
| McIntosh, 2019 | 44 | 9 | 47 | 0 | Unbiased (USA) |
| Sheets, 2018 | 23 | 59 | 19 | 0 | Unbiased (USA) |
| Stalsberg, 1989 | 17 | 0 | 0 | 83 | 2 accidents (NOR) |
| **SR-alone studies** | | | | | |
| Blancher, 2017 | 0 | 0 | 0 | 100 | 1 accident (FRA) |
| Eidenbenz, 2021 | 72 | 0 | 0 | 0 | Long burial (CHE) |
| Hatwal, 2021 | 0 | 92 | 0 | 8 | 1 accident (IND) |
| Kobek, 2016 | 100 | 0 | 0 | 0 | 1 accident (POL) |
| Pasquier, 2017 | 0 | 98 | 0 | 2 | Summer (CHE) |
| Tanaka, 2024 | 100 | 0 | 0 | 0 | Ski resort (JPN) |
| Zachau, 2020 | 29 | 62 | 0 | 0 | Media reports (NZ) |

* Only studies reporting the number or proportion of at least one activity preceding the accident are presented. Additional two studies [62,63] were excluded from the table since the samples were intentionally selected to represent recreationalists of a specific sport.
†Non-recreational
‡AUT: Austria; CAN: Canada; EU: Several European countries combined; ESP: Spain; FRA:France; IND: India; JPN: Japan; NOR: Norway; POL: Poland; CHE: Switzerland

or general), hunting, hiking, tramping, mountaineering, camping, scooting, group expeditions, playing, snowmobiling, working, driving, military exercises, and avalanche training. To better communicate the pattern, we further recoded the activities into with-ski (all recreational activities necessarily involving skis/snowboard), without-ski (all non-motorised recreational activities not necessarily involving skis/snowboard), motorised (all motorised recreational activities), and non-recreational (including working, driving, military exercise, avalanche control, and avalanche education). The results are illustrated in Table 3. Among unbiased sampling studies, 60% [22] to 100% [19,22,34,68,73,76,83] fatalities from avalanche accidents are recreationists, and considerable proportion of recreational activities were ski-related, ranging from 23% [76] to 100% [19,22,68].

Haegeli et al [32] noted Canadian sample had a greater proportion of people involved in mechanized backcountry skiing and snowmobiling, whereas the Swiss sample had a significantly greater proportion involved in nonmechanized backcountry and off-piste skiing. However, whether the role of such regional differences in activities could influence PCAD is not conclusive. Sheets et al. [76] examined the relationship between activities (including snowmobiling, backcountry touring, side country riding, inbound riding and climbing) and PCAD. They found incidence of trauma did not vary widely across these activities [76]. In contrast, by analyzing summer avalanche fatalities, Pasquier et al. [65] highlighted trauma became the leading cause of death in summertime, and 95% of trauma deaths occurred during alpine climbing. It is noteworthy that causes of avalanche deaths in summertime can be shaped by more seasonal features than activities alone, and more concrete evidence is needed for establishing the association. Zachau [61] examined avalanche deaths occurred in New Zealand in which climbers were highly over represented and found 61% were trauma. However, the study collected most of the data from media report, limiting the credibility of the PCAD estimate.

**Commercial connections.** We identified six studies [18,27,34,63,73,76] having potential conflict of interest with Black Diamond®, the producer of Avalung equipment. Since Avalung has been extensively marketed by highlighting asphyxia's highest proportion in causing avalanche death, we regard their results as being potentially biased towards higher proportion in asphyxia. The identification criterion was that a study mentioned Avalung by name, other than necessary cases such as objectively describing the equipment carried by victims; it was counted regardless whether the conflict of interest was stated [18,27,34,63,73,76]. We excluded these studies in the sensitivity analysis for commercial connection. (Note, this analysis was not planned in our study protocol, since we did not foresee such conflict of interest until we retrieved all studies.)

## Meta-analysis

The results in 24 cohorts for meta-analysis were pooled. The resulting estimates are shown in Fig 2A to 2C. Their pooled proportion estimates are 21% (CI 16–25%) for trauma, 80% (CI 74–85%) for asphyxia, and 2% (CI 1–4%) for hypothermia. For trauma, the observed proportions vary widely (PI 11–36%), and 41% of that variance reflects variance in true proportions ($I^2 = 43$%), indicating substantial heterogeneity. For asphyxia, the observed proportions vary widely (PI 54–93%), and 55% of that variance reflects variance in true proportions ($I^2 = 57$%), also indicating substantial heterogeneity. For hypothermia, while the observed proportions vary moderately (PI 0–9%), 0% of that variance reflects variance in true proportions ($I^2 = 0$%), indicating low heterogeneity.

**Risk of bias.** Most of the included meta-analysis studies have moderate risk of bias. The rating is available in S6 Tbale and illustrated in Fig 3.

**Table 3. Condensed table for subgroup analysis.**

| Cause* | Subgroup# | Ns, Nc† | Est(%)‡ | Heterogeneity§ | | | | p(power, %)‖ |
|---|---|---|---|---|---|---|---|---|
| | | | | PI(%) | Tau² | p | I²(%) | |
| **Time Span** | | | | | | | | |
| Trauma | 1970-2000 | 543, 8 | 23(20–27) | 19–28 | 0 | 0.66 | 0 | 0.07(13) |
| | after 2000 | 237, 6 | 29(21–39) | 16–46 | 0 | 0.42 | 0 | |
| | across 2000 | 650, 7 | 21(16–27) | 9–41 | 0.11 | 0.02 | 59 | |
| Asphyxia | 1970-2000 | 543, 8 | 72(68–75) | 67–76 | 0 | 0.26 | 21 | 0.05(39) |
| | after 2000 | 138, 6 | 82(72–88) | 58–94 | 0.11 | 0.32 | 15 | |
| | across 2000 | 487, 6 | 78(64–88) | 25–98 | 0.62 | <0.01 | 79 | |
| **Regions** | | | | | | | | |
| Trauma | Austria | 200, 4 | 14(5–34) | 0–95 | 0.85 | <0.01 | 78 | 0.45(12) |
| | Canada | 278, 4 | 23(19–29) | 14–36 | 0 | 0.82 | 0 | |
| | Norway | 57,2 | 18(10–30) | 10–30 | 0 | 0.39 | 0 | |
| | Japan | 189, 2 | 17(13–24) | 13–24 | 0 | 0.76 | 0 | |
| | USA | 250, 6 | 23(17–30) | 12–39 | 0.04 | 0.36 | 8 | |
| Asphyxia | Austria | 69, 3 | 90(80–95) | 5–100 | 0 | 0.25 | 27 | 0.07(23) |
| | Canada | 278, 4 | 74(69–79) | 61–83 | 0 | 0.29 | 50 | |
| | Norway | 57, 2 | 76(57-88) | 55–89 | 0.40 | 0.12 | 58 | |
| | Swiss | 56, 2 | 78(61–88) | 57–90 | 0.08 | 0.11 | 61 | |
| | USA | 250, 6 | 82(71–90) | 43–97 | 0.33 | 0.01 | 65 | |
| **Sample size** | | | | | | | | |
| Trauma | ≤30 | 226, 12 | 15(11–21) | 11–22 | 0.01 | 0.72 | 0 | 0.02(13) |
| | 31-75 | 250, 6 | 21(14–29) | 6–50 | 0.18 | 0.06 | 53 | |
| | >75 | 998, 5 | 26(21–30) | 13-44 | 0.05 | 0.01 | 68 | |
| Asphyxia | ≤30 | 226, 12 | 85(80–89) | 79–90 | 0 | 0.88 | 0 | <0.01(31) |
| | 31-75 | 282, 7 | 78(64–87) | 28–94 | 0.61 | <0.01 | 75 | |
| | >75 | 704, 3 | 71(68–74) | 46–88 | 0 | 0.23 | 31 | |
| Hypothermia | ≤30 | 198, 11 | 1(0–1) | 0–38 | 1.62 | 1 | 0 | 0.70(40) |
| | 31-75 | 250, 6 | 2(0–8) | 0–41 | 1.10 | 0.91 | 0 | |
| | >75 | 998, 5 | 2(1–4) | 1–8 | 0.09 | 0.15 | 40 | |
| **Data representativeness** | | | | | | | | |
| Trauma | Local | 513, 16 | 18(13–24) | 8–37 | 0.18 | 0.09 | 34 | <0.01(14) |
| | National | 440, 5 | 21(18–25) | 16–28 | 0 | 0.46 | 0 | |
| Asphyxia | Local | 513, 16 | 83(76-88) | 54–95 | 0.40 | <0.01 | 60 | 0.02(31) |
| | National | 277, 4 | 74(69–79) | 62–84 | 0 | 0.14 | 45 | |
| **Forensic diagnostic procedure** | | | | | | | | |
| Trauma | Mixed | 644, 9 | 25(20–31) | 15–40 | 0.06 | 0.07 | 45 | 0.02(13) |
| | Full or strategic | 100, 5 | 11(6–19) | 4–25 | 0 | 0.40 | 2 | |
| Asphyxia | Mixed | 545, 9 | 77(68–84) | 44–93 | 0.32 | <0.01 | 64 | <0.01(24) |
| | Full or strategic | 100, 5 | 81(79–92) | 72–95 | 0 | 0.43 | 0 | |

* Results for hypothermia are presented in S7 Table due to a lack of statistical power to detect hypothermia cases for most included studies, except for the sample size subgroup, where studies with sufficient power are selected into a subgroup (sample size > 75 subgroup).

# Predefined subgroups with <2 cohorts were removed from subgroup analysis. In forensic diagnostic procedure sub-group, "Mixed" means some cases underwent external examination alone, while others underwent either autopsy or both, due to NON-medical considerations such as refusal of fatalities' family; "Full or strategic" means either all cases underwent full autopsy (internal or internal+external autopsy) or they received either internal or external autopsy, according to medical considerations.

†Ns: Number of samples; Nc: Number of cohorts

‡Subgroup proportion estimates with 95% confidence interval. The proportions of three causes under one subgroup do not add up to 100%, because different studies are included for estimating each outcome, and sampling errors accounted for is partially dependent on the absolute number for each cause

§(a) PI is 95% prediction interval. (b) $p$ is $p$ for Cochran Q test.

‖ Cochran's Q statistic for subgroup differences and statistical power for the test.

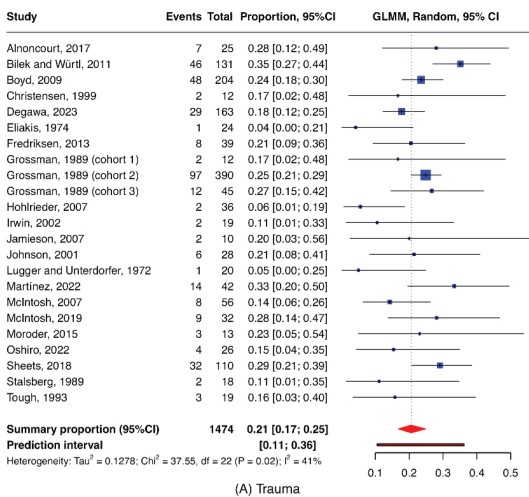

(A) Trauma

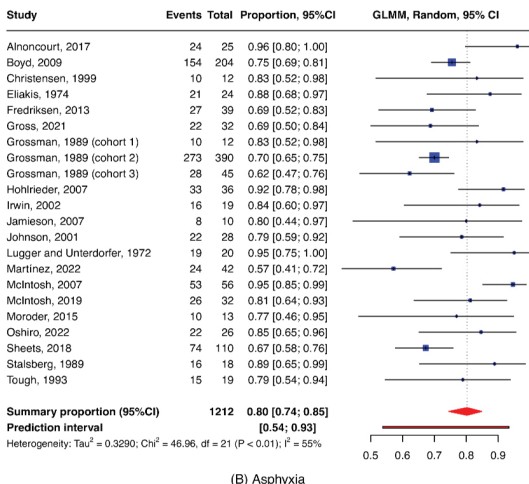

(B) Asphyxia

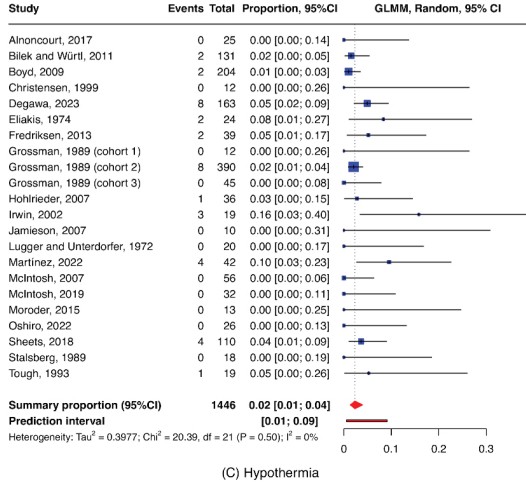

(C) Hypothermia

**Fig 2. Overall PCAD estimates for (A) Trauma, (B) Asphyxia and (C) Hypothermia.** *The size of blue squares visualizing point estimate of each study is adjusted by sample size. "Total" is sample size; "Events" is the number of corresponding causes.*

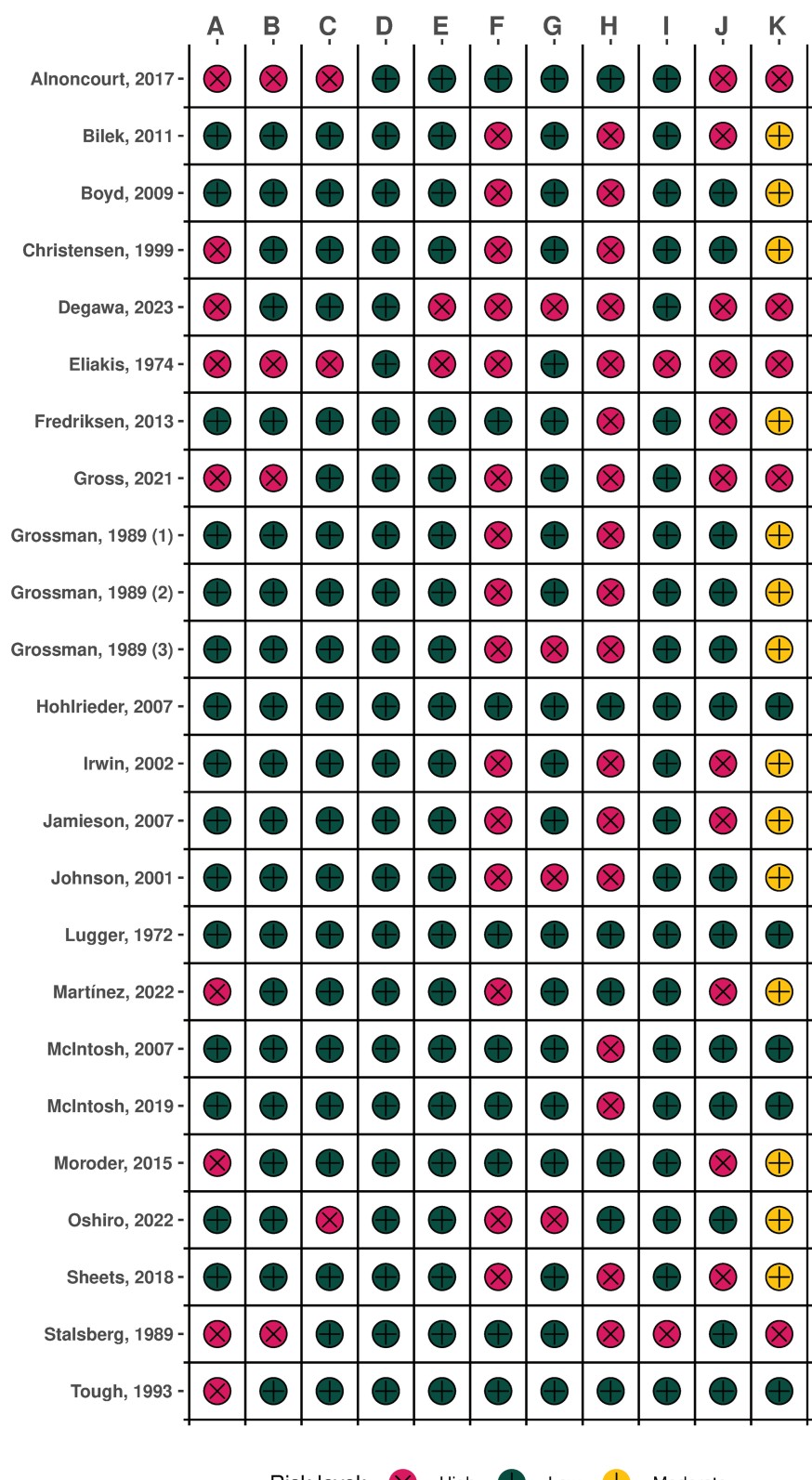

**Fig 3. Individual-study risk of bias.** *A.Representative target population; B.Sampling frame; C.Random selection; D.Non-response bias; E.Direct data collection from subjects; F.Case definition; G.Reliable and valid measurement; H. Mode of data collection; I. Length of the shortest prevalence period; J. Numerators and denominators; K. Summary.*

Among all the 24 cohorts, the summary risk of bias is high for five, moderate for 14, and low for five. For asphyxia, the two cohorts [69,77] that were not included had moderate and high risk of bias, respectively. For trauma, the one cohort [71] that was not included had a high risk of bias; for hypothermia, the two cohorts [34,71] that were not included have high and moderate risk of bias, respectively.

For details on the justification for the rating of each item, see S1 Text.

Notably, the items most often rated as 'high risk of bias' were: *case definition (F)* due to lack of forensic diagnostic criteria, followed by *mode of data collection (H)*, and *Numerators and denominators (J)*. In the end of S1 Text, we give more details on these items.

**Overlap.** Among included meta-analysis studies, we identified five cohorts risking overlaps. Two cohorts used USA data from 1992 to 1999 [34] and 1989 to 2006 [18], respectively. Here, overlap is not certain because they had different sources of data, and we hence included both in the original synthesis and removed the one [34] fully covered by another in sensitivity analysis for overlaps. Three cohorts used Canadian data in 1980 to 1991 [63], 1979 to 1985 [22], and 1984 to 2005 [26]. The overlap is again not certain because different data sources were used, so we included all in the original synthesis, and removed those with higher risk of bias [22,63] for sensitivity analysis.

**Sensitivity analysis.** The results of the *leave-one-out*, *risk-of-bias*, and *overlap* sensitivity analyses indicated that the pooled effect estimate remained stable across multiple iterations of the analysis. Removal of any single study (S2 Fig), overlapping group of studies (S3 Fig), or high risk of bias studies (S4 Fig) altered the overall effect sizes by 0% to 2% in trauma, 0% to 3% in asphyxia estimates, and 0% in hypothermia estimate. None of these changes were considered substantial. Similarly, the changes in heterogeneity indicators across these analyses were tiny. We planned to give special attention to Christensen et al. [62] and Tough et al. [63] considering their sample characteristics. By removing either of them, pooled proportion estimates and 95%CI remained unchanged.

In commercial connection sensitivity analysis (S5 Fig), trivial changes in pooled proportion estimates was detected for either trauma (22%, CI 18–26%, increased by 1% from 10%), asphyxia (78%, CI 72–84%, decreased by 2% from 80%), or hypothermia (2%, CI 1–4%, unchanged). Heterogeneity roughly remained at the same level with overall estimates.

**Subgroup analyses for exploring heterogeneity.** To explore the substantial heterogeneity in the proportions of trauma and asphyxia, we performed subgroup analyses. See Table 3. We selected the groupings based on findings in the meta-analysis phase.

*Time span.*

For trauma, the observed proportions in 1970–2000 and after-2000 subgroups vary to some extent (PI 19–28%, 16–46%, respectively). Notably, 0% of that variance reflect variances in true proportions ($I^2$ = 0% for both), indicating low heterogeneity. However, the observed proportions in across-2000 subgroup vary widely (PI 9–41%), and large portion of that variance reflects variances in true proportions ($I^2$ = 59%), suggesting substantial heterogeneity. This high heterogeneity in across-2000 subgroup alone further validates the role of time span in explaining heterogeneity of trauma. Compared to overall proportion estimate (21%), subgroup proportion estimates increase to 23% for 1970–2000 subgroup and to 29% for after-2000 subgroup, and the between-subgroup difference on proportion estimates approach statistical significance ($p$ = 0.07), and the statistical power is low (13%).

For asphyxia, the observed proportions in 1970–2000 and after-2000 subgroups vary moderately (PI 67–76%, 58–94%, respectively), and small to moderate portion of that variance is attributable to variances in true proportions ($I^2$ = 21% and 15%, respectively), indicating low to moderate heterogeneity. Meanwhile, across-2000 subgroup has large dispersion in

observed proportions (PI 25–98%), and a big portion of it can be attributable to variances in true proportions ($I^2$ = 79%), pointing to substantial heterogeneity. This again validates the role of time span in explaining heterogeneity of asphyxia. Compared to overall proportion estimate (80%), subgroup proportion estimates decrease to 78% for 1970–2000 subgroup and increase to 82% for after-2000 subgroup, and between-subgroup difference was significant on proportion estimates ($p$ = 0.05), but the statistical power is low (39%).

In addition, while $p$-value for Cochran Q was significant in overall estimate ($p$ = 0.02 for and <0.01 for asphyxia, see Fig 2A and Fig 2B), no difference can be revealed for 1970–2000 and after-2000 subgroups, either for trauma ($p$ = 0.66 and 0.42) or asphyxia ($p$ = 0.26 and 0.32), again suggesting the role of time periods in explaining heterogeneity. Consistent with this finding, for across-2000 subgroup, differences can be detected ($p$ =0.22 and <0.01).

*Regions.*
The findings show some regions have a meaningful degree of heterogeneity, while others not. For trauma, the observed proportions of Austria vary widely (PI 0–95%) with large ratio of variance attributable to true proportion ($I^2$ = 78%), suggesting substantial heterogeneity. United States vary, though still widely, in a much narrower range than Austria in observed proportions (PI 12–39%), and it has very small portion of variance reflecting variances in true proportions ($I^2$ = 8%), indicating moderate heterogeneity. Canada, Norway and Japan vary moderately in observed proportions (PI 14–36%, 10–30%, and 13–24%, respectively) with nil variance attributable to true proportion ($I^2$ = 0% for all), indicating low heterogeneity. Compared to overall proportion estimate (21%), subgroup proportion estimates decrease to 14% for Austria subgroup, and increase to 23% for Canada and USA subgroups.

For asphyxia, the observed proportions of Austria vary very widely (PI 50%–100%) with small ratio of variance attributable to true proportion ($I^2$ = 27%), suggesting substantial heterogeneity. Canada, Norway, Switzerland, and USA also vary widely (PI 61–83%, 55–89%, 57–90% and 43–97%, respectively), and they have large portion of variance reflecting variances in true proportions ($I^2 \geq$ 50% for all), indicating substantial heterogeneity. Comparing to overall proportion estimate (80%), subgroup proportion estimates increase to 90% for Austria subgroup.

The result for Norway and Japan should be interpreted with caution, since the subgroup only includes a small number of two cohorts. Moreover, three of the four studies in Canada subgroup used national representative data, and we therefore cannot exclude that the heterogeneity is explained by data representativeness (see section below).

*Sample size.*
Reduced heterogeneity was observed in n≤30 subgroups for both trauma and asphyxia: their observed proportions vary in a narrow range (PI 11–22% and 79–90%, respectively) with nil amount of variance reflecting variance of true proportion ($I^2$ = 0% for both).

Hypothermia already shows low heterogeneity (high dispersion but none of the variance is from variances of true proportion) in overall proportion estimate. However, meta-analysis with extremely small pooled estimates (defined as <10%) often associates with low $I^2$ value regardless of the true heterogeneity [89]. Therefore, we continue to look for possible moderators of hypothermia to reduce its between-study variation of observed proportions. We observed in sample size >75 subgroup, though the proportion estimate remains unchanged from overall estimate, variations of observed proportions across studies reduced (PI for the subgroup estimate: 0.01–0.08, PI for overall estimate: 0–0.09) and dispersion no longer covers zero.

*Data representativeness.*

Reduced heterogeneity was observed in national representative subgroups for trauma (PI 16–28%, $I^2 = 0$) and asphyxia (PI 62–84%, $I^2 = 45\%$): the observed proportions vary moderately (PI 0.62–0.84) and 45% of that variance reflects variances in true proportions ($I^2 = 45\%$). In contrast, local representative subgroups' heterogeneity remained substantial for trauma (PI 8–37%, $I^2 = 34$) and asphyxia (PI 54–95%, $I^2 = 60\%$). The findings can be further evidenced by insignificant $p$ of Cochran's Q within each national subgroup (0.46 for trauma; 0.14 for asphyxia) and significant or approximately significant $p$ of Cochran's Q within each local subgroup (0.09 for trauma; <0.01 for asphyxia ). Furthermore, $p$ of Cochran's Q between local and national subgroups was significant (<0.01 for trauma; 0.02 for asphyxia).

*Forensic diagnostic procedure.*

Heterogeneity reduced to low levels in full/strategic-autopsy subgroup for both trauma (PI 4–25%, $I^2 = 2\%$) and asphyxia (PI 72–95%, $I^2 = 0\%$). In contrast, mixed-autopsy subgroup heterogeneity remained substantial for trauma (PI 15–40%, $I^2 = 45$) and asphyxia (PI 44–93%, $I^2 = 64\%$). The findings can be further evidenced by insignificant $p$ of Cochran's Q within each full/strategic-autopsy subgroup (0.40 for trauma; 0.43 for asphyxia) and significant or approximately significant $p$ of Cochran's Q within each mixed-autopsy subgroup (0.07 for trauma; <0.01 for asphyxia ). Furthermore, $p$ of Cochran's Q between two subgroups was significant (0.02 for trauma; <0.01 for asphyxia).

It is noteworthy that, comparing to overall proportion estimates (21% (PI 11–36%) for trauma; 80% (54–93%) for asphyxia), the pooled proportion estimates from full/strategic autopsy subgroup shifted towards lower estimates noticeably (11% (PI 4–25%)) for trauma, and towards high estimates slightly for asphyxia (81% (72–95%)).

**External validity.** Compared with the unbiased estimates, the hypothermia sample (all fatalities having macroscopic signs of hypothermia) [12] differed significantly in all proportions of cause; the long-burial sample [33] differed significantly in asphyxia ($\chi^2(1) = 4.11$, $p=0.04$) and hypothermia ($\chi^2(1) = 25.43$, $p<0.01$); the summertime sample [65] differed significantly in trauma ($\chi^2(1) = 92.36$, $p<0.01$) and asphyxia ($\chi^2(1) = 68.81$, $p<0.01$). The results are summarised in Table 4.

## Discussion

### Summary of findings

Our work presents a systematic review of influencing factors to PCAD on the full set of 37 studies. This review suggests four results. First, burial time and depth are well-established influencing factors to PCAD, but they are often reported with insufficient detail. Second, there is scarce but strong evidence supporting the association of PCAD and snow climates. Third, there is only anecdotal evidence on the association of PCAD with time periods and location of the accidents, representativeness of the data, fatalities' age, expertise, use of mitigation gear, and medical practitioners' forensic procedures. Fourth, existing evidence on association of PCAD with terrain forms is conflicting. Fifth, recreational on-ski sports are the most prevalent activities preceding an avalanche accidents.

Furthermore, our work presents a meta-analysis of evidence from 22 studies (24 cohorts) using an unbiased sampling framework. Our analyses suggest the following eight results. First, the highest proportion of avalanche deaths are due to asphyxia (81% (CI 74–85%)), followed by trauma (21% (CI 17–25%)), and then hypothermia (2% (CI 1%–4%) (the proportions do not add up to 100%, because different studies are included for estimating each outcome, and sampling errors accounted for is partially dependent on the absolute number for each cause).

**Table 4. Condensed table for external validity tests.**

| Study feature* | Causes | Biased† vs unbiased‡ | Test result§ |
|---|---|---|---|
| Hypothermia [12] | Trauma | 0 vs 23% | $\chi^2$ (1) = 4.74, p = 0.03 |
| | Asphyxia | 50% vs 72% | $\chi^2$ (1) = 8.09, p <0.01 |
| | Hypothermia | 50% vs 2% | $\chi^2$ (1) = 116.72, p <0.01 |
| Long-burial [33] | Trauma | 13% vs 21% | $\chi^2$ (1) = 2.51, p= 0.11 |
| | Asphyxia | 61% vs 78% | $\chi^2$ (1) = 4.11, p= 0.04 |
| | Hypothermia | 13% vs 2% | $\chi^2$ (1) = 25.43, p <0.01 |
| Summertime [65] | Trauma | 19% vs 21% | $\chi^2$ (1)= 92.36, p <0.01 |
| | Asphyxia | 61% vs 78% | $\chi^2$ (1)= 68.81, p <0.01 |
| | Hypothermia | 0 vs 2% | $\chi^2$ (1)= 0.86, p=0.353 |

* Proportion of each biased sample was compared with the summary proportion obtained in the current meta-analysis
†*a.* hypothermia sample: all fatalities having signs of hypothermia; *b.* Long-burial sample: all fatalities buried for > 60 min; *c.* Full burial: All fatalities who underwent full burial; *d.* summer sample: all non-survivors being caught in summertime
‡for trauma and asphyxia, pooled proportion estimates from 1970-2000 subgroup were used for comparing with observed proportions from "hypothermia" sample [12], which used data between 1980 an 1987; pooled proportion estimates from across-2000 subgroup were use for comparing with observed proportions from "long-burial" sample [33] (data time periods: 1997–2018) and "summertime" sample [65] (data time periods: 1984–2014). For hypothermia, pooled proportion estimate for ≤ 75 subgroup was used.
§Results with significant differences were bold.

Second, no individual study disproportionately affects estimated proportions, establishing the credibility of the results. Third, no evidence was found for a role of potential data overlaps or commercial connections on estimated proportions. Fourth, the observed proportions for death by trauma and asphyxia vary substantially across cohorts, pointing to the importance of exploring the reasons of these variations. Fifth, time period of accidents explained the between-study variations for trauma and asphyxia to a fair extent, and sample size explained the between-study variations for hypothermia to a fair extent. Sixth, for the period 1970 to 2000 the proportion of death by trauma is 23% (CI 20%–27%) and that by asphyxia is 72% (CI 68%–75%), whereas for the period after 2000 trauma is 29% (CI 21%–39%) and asphyxia 82% (CI 72%–88%); proportion of death by hypothermia is 2% (CI 1%–4%) for n>75 subgroup. Seventh, nationally representative data and full/strategic autopsy leading to the forensic diagnosis explained the between-study variations for trauma and asphyxia to a fair extent. Eighth, more than half of the studies included for meta-synthesis have high risk of bias, with a lack of reporting forensic diagnostic criteria being the most prevalent issue.

## Strengths of the study

We aimed for methodological rigor by selectively using 22 of 37 studies with unbiased sample for our meta-analysis, which is crucial for accurate estimations of avalanche death causes. This approach enhances reliability by excluding any bias potentially found in studies of specific factors or single events. Meanwhile, the systematic review included all studies, offering a broad overview and deep insight into the factors influencing PCAD. We believe this strategic bifurcation provides a clearer understanding of the domain.

The meta-analysis encompasses a substantially larger dataset than previous reviews [11,14, 25,57]. This extensive aggregation provides robust statistical power and improves the generalizability of our findings across different geographic and situational contexts. This allows our work to offer a more comprehensive overview of PCAD research.

The work is the first study on PCAD using a GLMM-based meta-synthesis approach [37], enhancing the precision and reliability of our estimates by accounting for both within-study and between-study uncertainties. Adopting this method is crucial in the context of avalanche fatality research, which often deals with small sample sizes (thanks to intrinsic low probability of avalanche death) and rare events (such as hypothermia). Unlike previous studies that relied on naïve pooling or narrative summary, our approach adjusts for variability and differential quality among studies, thus avoiding the over-representation of smaller or potentially less-rigorous studies.

This work adopted a multi-faceted strategy of heterogeneity assessment that combines $I^2$ and PI proposed by Borenstein *et al.* [90]. Using $I^2$ statistics alone is the most common practice in meta-analysis, but is also heavily-questioned [90–92]. There are also other indicators for heterogeneity assessment, including $Tau^2$ and Cochran's Q statistics. However, multiple methodological studies support PI as a robust heterogeneity assessment [37,91,93]. Under relevant conditions, PI covaries with $Tau^2$ (PI length is approximately $4 \times Tau$ if within-study variances are small [44]), yet provides a more intuitive interpretation. Whereas, the power of Cochran's Q to detect heterogeneity is limited for smaller sample sizes (*n* of included studies). Therefore, we reported but treated as auxiliary references these indices of heterogeneity. This approach enhances the reliability of our findings, especially from subgroup analyses. To our best knowledge, our work is one of the first meta-analyses of prevalence to adopt multiple heterogeneity indicators, with special emphasis given to PI.

## Comparison with other studies

**Proportions of asphyxia, trauma, and hypothermia in general forensic autopsy.** Our findings reveal that asphyxia is the predominant cause of avalanche deaths, occurring in 80% (CI 74-85%) of cases, which is significantly higher compared to its prevalence in general forensic autopsies (denominator is all-cause death rather than avalanche death) where asphyxial deaths account for only 4.9% [94] to 15.7% [95]. This substantial difference underscores the unique lethal risks posed by avalanche burial. Similarly, the proportion of trauma-related deaths in our study (21%, CI 17-25%) exceeds the 8% prevalence reported in general forensic contexts [96], highlighting the severe impact of physical injuries in avalanche incidents. The prevalence of hypothermia as a cause of death in avalanche cases (2%, CI 1-4%) remains relatively low, yet it is notably higher than in the general population, where it rarely exceeds 0.5% [97]. These disparities emphasize the specific coldness hazards associated with avalanches.

**Other PCAD reviews.** Previous reviews have focused on either naïve pooling or narrative summary of proportions of causes of avalanche death.

Stalsberg *et al.* [11] combined their finding on PCAD with four existing studies at the time [19,30,56,70] by naïve pooling (unweighted additions of cases from multiple studies), and concluded with 69.1% for asphyxia (including drowning), 2.9% for a combination of asphyxia and trauma, 13.2% for trauma, 3.7% for hypothermia, and 11% for cases with unknown causes. The results for trauma and asphyxia are less than the lower end of the 95% confidence intervals in our study. Notably, Stalsberg and colleagues' proportions can be heavily under-estimated, considering the 11% cases with unknown forensic diagnosis. Furthermore, due to the weakness of naïve pooling, small sample studies and methodologically-weaker outlier studies can be over-represented in the finding.

Pasquier *et al.* [25] reported the minimum and maximum observed PCAD across the included studies on proportions of causes of avalanche death: 65-100% for asphyxia, 5–29%

for trauma and 0–4% for hypothermia. In contrast, our finding determined a narrower 95% confidence interval for trauma and asphyxia. This reflects a less uncertain estimation that accounts for study variability, in addition to further confirming the reliability of our results. Our findings for hypothermia (CI 1-4%) align closely with their results and consolidate its lower prevalence as a cause of avalanche deaths.

## Trends of PCAD over time

There is a long-standing conjecture suggesting shifts in the PCAD [57], perhaps thanks to improved information on snow conditions and avalanche situations [98], mitigation equipment [71,76,98], avalanche education/awareness [73], self-rescue skills [26], time to extricate [74,76,79], on-site triage tools [78,99] and treatment guidelines [25], bystander CPR skills of back-country recreationists [74], meteorological variations such as changes in snow density over time [71,100,101]. Our subgroup analysis shows slight but insignificant difference for PCAD between the periods, reflected by substantial overlap of 95%CI of the two estimates: for trauma, we noted a proportion of 23% (CI 20–27%) in the 1970-2000 subgroup, which turned to 29% (CI 21–39%) in the after-2000 subgroup, while asphyxia showed a change from 72% (CI 68–75%) to 82% (CI 72–88%) across the same periods. However, this subgroup strategy accounts for substantial amount of heterogeneity from overall proportion estimates (according to none to low $I^2$ and Cochran's Q test), denoting time's role in explaining between-study variations. Although the significant or approximately significant differences between these subgroups strengthen the validity of the finding, caution should be exerted in interpreting the differences due to insufficient statistical power (13% for trauma and 39% for asphyxia). Note that in the study we estimated proportions, therefore these changes point to relative composition of different causes by death, rather than absolute change in number of fatalities (which was stable in Europe [28], Canada [100] and United States [100]).

A subjective and widespread impression is the proportion of traumatic death should be stable, considering its inevitability [57] and the fact that preventive equipment such as airbag packs may not have prevented traumatic fatalities [76]. We find no contradictory evidence: while the subgroup **point** estimates for trauma increased over time (from 23% to 29% for trauma), the dispersion of observed values reveals no clear direction of the changes of these proportions (PI of 1970–2000 subgroup is covered by the range of after-2000 subgroup). This suggests that the between-study variation explained is associated with more diverse patterns within the post-2000 subgroup. We consider several possibilities for the wider observed proportion dispersion across after-2000 subgroup studies: backcountry skiers' increasing but varying rate of wearing a helmet, improving and/or changing practices in forensic differentiation criteria, or both. Indeed, lethal head injuries accounted for 28% [34] to 31% [76] of traumatic avalanche death, for which a helmet can be an effective prevention [24]. The possible role of forensic differentiation will be discussed in one of the sections below.

Asphyxiation takes some time to be lethal and usually precedes the stage of hypothermia, allowing more time for rescue compared to mechanical injuries. Therefore, many recent improvements in avalanche survival techniques aim to prevent asphyxiation, including prevention of burial, facilitating extrication of buried victims, and improving resuscitation of extricated buried victims [57,102]. However, we observed an change in **point** estimates of PCAD by asphyxia over time (from 72% to 82%), and the dispersion of observed proportion also moved to higher side mildly (PI 68% to 75% for 1970–2000, and 72%–88% for after-2000 subgroups). The extrication of buried victims, who are more susceptible to asphyxia, adds to

difficulty and uncertainty of successful rescue of fatality having higher risk of asphyxia. This could lead to a slower improvement in saving asphyxial than traumatic victims. This suggests studies with focused research targets need to be undertaken to find out the net effectiveness of each procedure.

## Regional difference of PCAD

It has been pointed out that regional factors can lead to PCAD discrepancies, at macro- and micro-regional scales [18,57]. Frequently considered macro-regional factors include snow avalanche climate [79], adequacy of emergency medical responses [80], adoption of different safety measures [71], and popularity of different types of winter sports [26,73]. Micro-regional factors (conditions of obstacles in avalanche path) include above/below treeline [18,26], density of forestation [26], and presence of cliff bands [18]. To assess the matter, we investigate potential regional differences with two sub-grouping strategies: first, by country where the accidents occurred, and second, by the representativeness of the accident data (local/national). However, the country subgroup does not reveal meaningful and conclusive within-country similarity. This is consistent with another empirical evidence against the role of micro-regional factors, where the authors compared the trauma-related accidents above and below tree line [87]. The study reveals that avalanches occurring in forested terrain do not elevate the risk of traumatic fatalities [87]. Sheets et al. [76] speculated that the lower rates of trauma in the European studies may reflect the different research methods and lack of autopsy data more than terrain or activity. Another plausible interpretation of our result is stratification on country-level does not provide sufficient micro-regional granularity that allows for detecting any pattern.

However, our finding from the data representativeness subgroup suggests that macro-regional effects override micro-regional effects' role in shaping PCAD by asphyxia. We find nation-representative cohorts, when pooled with each other regardless of the country, led to considerably reduced heterogeneity for both trauma (PI 16–28%, $I^2 = 0$) and asphyxia(PI 62–84%, $I^2 = 45\%$), while local representative cohorts led to an increase. And this finding has low risk of being confounded by the bigger sample sizes of national representation studies, since in analysis of larger sample-size subgroups (sample size 31–75 and > 75), heterogeneity for asphyxia was not as well explained as in the national representation subgroup. This new possibility is further substantiated by survival curves, which have similar four-phase shape despite sourcing from accident data of different countries [30–32].

## Roles of burial-related factors to PCAD

Burial-related parameters are well-recognized factors influencing PCAD [64]; however, their subgroup was precluded due to their sparse reporting in the studies analyzed. Alongside establishing external validity of the estimates from meta-analysis, the external validity test also addresses this limitation. The tests compared overall proportion estimates from meta-analysis with results from studies focusing on fatalities involving full and long burial (>60 mins). This comparison not only validated our findings but also reinforced the established evidence that burial characteristics significantly influence the distribution of causes in avalanche-related death.

Specifically, we find fatalities who underwent full and long burial die six times more often due to hypothermia than unbiased avalanche fatalities, reflected by pooled proportion estimates. Meanwhile, proportion of traumatic death stays the same, suggesting burial time (in case of full burial) shifts the causes of avalanche death from asphyxia towards

hypothermia. This finding replicates existing evidence and the intuitive logic of burial type and burial duration's role in shaping PCAD [32,64,101]. It is also consistent with recent research analyzing 1,643 critically buried individuals in Switzerland, which found that survival chances were approximately three times higher for those rescued by companions compared to those relying on organized rescue teams [101]. The authors partly attributed the improvement to a decrease in median companion rescue time from 15 to 10 minutes, aligning with the critical 10-minute asphyxia threshold that significantly impacts survival. These findings underscore the critical importance of self and companion rescue, as well as the need for the rapid dispatch of organized rescue teams to improve survival outcomes.

The survival rate for individuals rescued by companions improved from 68.0% (134 of 197) between 1981 and 1990 to 74.8% (604 of 808) between 1981 and 2020 ($\chi^2$ test $p = 0.07$).

We also find fatalities who were caught by summer avalanche, when compared with unbiased avalanche fatalities from our meta-synthesized proportion estimates, never die of hypothermia and very much less often die of asphyxia, but more often die of trauma. Avalanche burial is legitimately shallower in summertime, and most of summer avalanche accidents occur during hiking or climbing, with the accident party falling over rock. Therefore, this finding suggests burial depth plus mechanical injuries shift the causes of avalanche death from asphyxia and hypothermia towards trauma, which is consistent with intuition and existing evidence [32,64]. However, current data does not allow us to extract the net effect of shallower burial.

## Forensic diagnostic procedures of PCAD studies: An issue plaguing credibility

Autopsies are crucial for confirming the cause of death in avalanche fatalities. The importance for PCAD studies has been highlighted [14,62,71,76], and it has been argued that autopsy is the only way to differentiate hypothermia from asphyxia [68]. Medical examiners' decisions to perform an external or internal autopsy on avalanche fatalities were based on circumstances of death, potential legal liability, and corpse appearance on external examination [73]. Brugger *et al.* pointed out that full external forensic examination should be deemed as the minimum from which to draw postmortem conclusions for a PCAD investigation [14]. Our meta-analysis provides empirical evidence supporting these viewpoints: pooled estimates based on studies with cause of death diagnosis obtaining through full/strategic autopsy shows substantially reduced heterogeneity for PCAD by trauma and asphyxia, comparing to a mixture of internal and external autopsy due to non-medical considerations. Furthermore, we observed a noticeable decrease of PCAD by trauma in the subgroup, suggesting PCAD by trauma can be over-estimated without proper forensic procedures.

Meanwhile, the forensic issue raises challenges for PCAD studies in many aspects: autopsy for avalanche fatalities is not a default in some countries such as Japan [75]; autopsy may be refused by the family members of the fatalities [11]; forensic diagnoses given by different medical examiners/coroners may have varying diagnostic bases, even within the same register [83]; differentiation diagnosis is sometimes difficult and mistakes can be made (e.g. asphyxia may be mistakenly assumed to be the cause of death because fatal internal traumatic hemorrhage may have only minimal external evidence [76]); forensic diagnostic criteria are evolving (e.g. a reliable histological criterion for the diagnosis of death by hypothermia was

not discovered until 2004 [68]); and in the scientific community of forensic medicine, the definition and sub-categorization of asphyxia have been inconsistent and controversial [17,103–105].

Given the significant variability in autopsy practices and the diagnostic challenges highlighted, it is imperative for researchers conducting PCAD studies to meticulously document the forensic methods that lead to the forensic diagnoses adopted in their study. The goodness of these practices can be well delineated by items F (case definition), H (mode of data collection), and J (numerators and denominators) in our risk of bias rating. However, only one [19] out of the 24 cohorts for our meta-analysis was rated as low risk for all the three items, suggesting tremendous under-reporting of forensic practices, complicated further by the inherent ambiguity and complexity in the forms of asphyxia.

The ambiguity stems from the term 'asphyxia' being too broad for a precise forensic diagnosis. While most of PCAD studies used "asphyxia" as a death diagnosis, the term does not always assist in delineating the underlying lethal mechanisms [104]. In case of avalanche death, the aetiology of asphyxia can be: *a.* hypoxic due to failure of supply (such as deprivation of oxygen underneath snow [18,106]); *b.* hypoxic due to failure of transfer (such as snow aerosol [30] or food remnants [11,106] aspiration); *c.* mechanical due to trauma (such as chest compressed by heavy snow burial [11,105]); or *d.* mechanical due to trapped position (such as respiration impeded by position of buried body [107]). Differentiating, at minimum, between hypoxic and mechanical aetiologies of asphyxia can be informative. However, selecting a terminology or defining detailed subcategories is not enough. At least three different terms are commonly used to describe the same mechanical aetiology, including traumatic asphyxia, crush asphyxia, and compression asphyxia [106]. Although considered interchangeable, the terms are subtly different, principally in relation to the circumstances in which the patient becomes compromised [106]. Moreover, there are at least six variations to this categorization and a standardization in the forensic medicine community is still lacking [17]. Clarity can only be achieved by meticulously documenting the forensic diagnostic criteria used in the study.

The complexity lies in the difficulty in distinction between asphyxia and trauma-related deaths [103]. Asphyxia with a mechanical origin may present with severe, fatal injuries, such as rib or cervical fractures, which can lead to confusion with trauma-related deaths. Furthermore, evidence suggests that many postmortem signs typically associated with asphyxial deaths lack specificity [103]. Upon reviewing the Canadian case records of avalanche accidents, we identified a specific incident from 1999 where a fatality was reported as "asphyxiated due to compression of his chest" [66]. However, a study included in our meta-analysis, which analyzed avalanche accidents from 1984 to 2005 using the same data source, categorized forensic diagnoses solely as asphyxia, trauma, or hypothermia [26]. It is possible that this particular case was classified within that study under 'asphyxia', or under trauma due to his chest injuries. However, the study failed to disclose the forensic diagnostic criteria, which can be disorienting for readers.

Failure to achieve this clarity in PCAD studies can be consequential. In terms of imprecise diagnoses due to unclarified ambiguity, it can potentially mislead the development and selection of avalanche mitigation equipment. This is because hypoxic and mechanical aetiology can have different mechanism and hence different mitigation strategies. For instance, the effectiveness of devices providing supplemental air during an avalanche burial [108] might be overestimated if they are assumed to be equally beneficial for both mechanical and hypoxic asphyxia. Gary [109] described a case where a woman, completely buried by an avalanche, could not benefit from receiving mouth-to-mouth resuscitation due to chest compression from the snow. Resuscitation finally became feasible after her chest was freed. Thus, oxygen

supply alone might not prevent asphyxia under physical compression. Further supporting this, Procter *et al.* [64] analyzed 633 fully buried avalanche victims, finding that deep burial (greater than 120 cm) independently increased mortality nearly fivefold compared to burials shallower than 40 cm, after accounting for burial duration. The critical factor here may be the weight on the victim's chest preventing exhalation. Clear forensic diagnostic criterion in PCAD is necessary for us to dig into this issue.

On the other hand, unclear classification due to unclarified complexity also has significant implications. It compels us to reconsider some prevalent interpretations of PCAD findings. For instance, traditionally, the higher proportion of fatal trauma in Western Canada compared to Europe has been attributed to forested terrain, which increases the likelihood of fatalities colliding with trees. However, linking high fatal trauma to forestation is inconsistent with empirical evidence of the effect of forestation on trauma. Using an estimated tree line elevation of 3500 meters, no increase in trauma-related incidents at elevations below this threshold was found [87]. By recognizing the potential for mechanical asphyxia being categorized under 'trauma', it is plausible to consider that Western Canada's elevated risk of trauma might also stem from high proportion of mechanical asphyxia. Because of its proximity to the Pacific Ocean, the influence of prevailing westerly winds, and orographic lift from the coastal mountain ranges, the western coast of Canada receives significantly more precipitation than the Alps [110], leading to denser snowpacks [32]. This increased snow density can exert greater pressure on buried victims, enhancing the probability of traumatic compression. If the forensic diagnostic criteria were clear for the studies included for our meta-analysis, the proposition could be immediately substantiated or falsified.

Consequently, we advocate for best practices in studying and reporting proportions of causes of avalanche deaths, that emphasize the thorough investigation and detailed disclosure of diagnostic criteria. This approach enhances the clarity and reliability of research findings. In instances where diagnostic criteria remain unspecified in our included studies for meta-analysis, a suboptimal yet somewhat more informative method involves categorizing causes of death as 'traumatic asphyxia' or 'trauma and asphyxia', adopted by five cohorts [18, 19,67,68,73]. This terminology serves to differentiate these conditions from hypoxic asphyxia. However, this practice still retains some ambiguity, particularly with the use of 'trauma and asphyxia', which may can be confused as meaning 'hypoxia asphyxia concomitant with other fatal injuries'.

## Implication

This work reaffirms asphyxia as the predominant cause of avalanche deaths, followed by trauma, and hypothermia. While our findings do not contradict traditional knowledge, they enhance it by providing more precise range estimates for these causes. We believed this improved precision supports on-site treatment and prevention guidelines with higher quality evidence and will facilitate evidence-based prioritization of medical resources, development of rescue equipment, and policy and protocol development. More importantly, based on these findings, we further revealed the role of time period in explaining variations in the proportions, indicating possible changes in avalanche dynamics and also enhanced but regionally discrepant medical, avalanche education, and risk mitigation practices. Additionally, the majority of fatalities were found to be associated with ski-related sports, emphasizing the need for targeted safety measures in these activities.

A critical finding from our study is the substantial under-reporting of forensic diagnosis criteria within the included studies. This implies that the credibility of the results of PCAD

studies is compromised, making it difficult to fully trust the reported proportion of causes of death. Also, under-reporting contributes to increased variability across studies, complicating efforts to synthesize and compare findings effectively. As a result, some established understandings of PCAD become questionable, potentially misleading the development and selection of risk mitigation equipment. Future studies on PCAD must stick to better reporting practice in forensic diagnosis criteria.

Furthermore, although we tried supplementing our search by using different languages, we did not find quality studies reporting PCAD of developing countries, except for one from India. This highlights a significant gap in research from developing countries. As backcountry skiing gains popularity in nations such as China, Turkey, and Chile, there is a crucial need to address the disparity in the quality of information sharing, avalanche danger forecasting, and emergency rescue services. Tourists and winter-sport enthusiasts seeking the thrill of powder snow in these regions may not have access to the same level of prepared resources, trained rescuers, and established clinical pathways as found in more traditionally frequented areas.

## Limitations of the study

Even though we did not set constraints on language in searching for relevant studies, our search strategy is in English and is not sensitive to non-English studies lacking English titles and/or abstracts. And there can be databases archiving studies for specific non-English languages that we do not cover. It was not within our resources to identify all possible databases and search them with validated translations. We addressed this by asking domain experts with multiple language backgrounds to suggest studies and data sources that we failed to cover.

In our study, the proportion estimates of three causes do not add up to 100%. This is because different studies are included for estimating each outcome; the sampling errors accounted for are partially dependent on the absolute number for each cause; and each proportion is estimated separately. Trikalinos et al. [111] suggest using a multi-nominal model that enforces a constraint in estimating the pooled proportion that the sum of the pooled proportions should be 100%. However, this model assumes mutual exclusiveness of the outcomes. Our included studies vary in whether they reported all three causes, two causes, or a combination of multiple causes, which prevented us from adopting this approach. This also undermines the validity of normalizing the sum of the three estimates to 100%. Importantly, we believe this does not invalidate our proportion estimates and between-subgroup comparisons. Separate analysis without imposing artificial trade-offs between causes can account for heterogeneity of each cause more properly (especially when there are different number of studies for each cause). For this reason, our practice better fits one of the principal aims of the study, which is to explore the factor that explains the high heterogeneity in the overall proportion of estimates. Separately estimating each correlated outcome is also the practice of other meta-analyzes with a design similar to ours [112,113]. As suggested by Barker et al. [114], for proportional meta-analyses, differences between subgroups can be compared and contrasted using a statistical test. This practice is also used in other meta-analyzes with a design similar to ours [113]. In our study, the various categorizations of causes have been fully documented and the implications have been accounted for by our risk-of-bias analysis and fully discussed. Our subgroup comparisons were always made by referring to both the proportion estimates and statistical indicators (CI, PI, Cohrane's Q, etc.)

We are unable to conduct subgroup analyses on burial type, depth, and duration, and their impact on the proportions of causes of avalanche deaths, due to the lack of reporting

of these features in included studies. In addition, some subgroups were now inspired by the meta-analysis. However, different types of groupings could be used in the future work.

We computed a minimal sample size of 75 for estimating hypothermia. Although reduced heterogeneity is achieved with the cutoff in sample-size subgroup analysis, we did not have any studies with sample size ranging from 75 to 110 in n>75 subgroup (actual range of sample size: 110–204).

We did not include analysis of publication bias, because PCAD studies report all fatal avalanche incidents rather than outcomes based on statistical significance. Therefore, it is unlikely to miss studies due to it.

Finally, the present work is based on synthesizing studies mostly with moderate to high risk of bias, limiting the reliability of its result.

## Conclusions

We re-affirm asphyxia as the predominant cause of avalanche deaths, followed by trauma, and then hypothermia. Patterns of PCAD by trauma and asphyxia became more diverse after the year of 2000. A sample size > 75 is needed to estimate the proportion of hypothermia. Regional PCAD discrepancies can be reduced by using nationally representative samples. Without proper forensic diagnosis procedure, PCAD by trauma can be overestimated. The results of meta-analysis build upon synthesizing and summarizing studies mostly with moderate to high risk of bias and should be interpreted with caution. Under-reporting of forensic diagnostic criteria is an important bottleneck to the reliability of evidence in the field. Existing evidence on the role of other influencing factors to PCAD such as fatalities' expertise and usage of mitigation gears is anecdotal and entails further research.

## Supporting information

**S1 Fig. Avalanche survival curve first proposed by De Quervain in 1966.**
(TIF)

**S1 Table. PRISMA 2020 Checklist.**
(DOCX)

**S2 Table. Searching strategy.**
(PDF)

**S1 Text. Risk of bias tool and summary of key items.**
(PDF)

**S3 Table. Author contacting effort and response.**
(PDF)

**S4 Table. Data screening history.**
(PDF)

**S5 Table. Table of extracted data.**
(PDF)

**S2 Fig. Forest plots for LOO sensitivity analysis.**
(TIF)

**S3 Fig. Forest plots for overlap sensitivity analysis.**
(TIF)

**S4 Fig. Forest plots for high risk-of-bias sensitivity analysis.**
(TIF)

**S5 Fig. Forest plots for commercial connection sensitivity analysis.**
(TIF)

**S6 Table. Ratings of risk-of-bias assessment.**
(PDF)

**S7 Table. Subgroup analysis for hypothermia.**
(PDF)

## Acknowledgments

We would like to express our sincere gratitude to Dr. Frank Techel from WSL Institute for Snow and Avalanche Research SLF for providing invaluable feedback on the first manuscript. His insightful comments and suggestions greatly improved the quality of our work. We also wish to thank geographer He Yujing for her verification of geographical information and Dr. Nina Scholten, Dr. Natalia Postnova and Dr. Andrey Rodionov for interpreting studies written in some non-English languages. Their help was crucial for the development of this study.

## Author contributions

**Conceptualization:** guang rong, Lauri Ahonen, Benjamin Ultan Cowley.

**Data curation:** guang rong, Lauri Ahonen, Gerit Pfuhl.

**Formal analysis:** guang rong, Lauri Ahonen.

**Investigation:** Lauri Ahonen.

**Methodology:** guang rong, Lauri Ahonen, Gerit Pfuhl, Benjamin Ultan Cowley.

**Project administration:** Lauri Ahonen, Benjamin Ultan Cowley.

**Resources:** Lauri Ahonen, Gerit Pfuhl, Benjamin Ultan Cowley.

**Software:** guang rong, Lauri Ahonen, Benjamin Ultan Cowley.

**Supervision:** Lauri Ahonen, Gerit Pfuhl, Benjamin Ultan Cowley.

**Validation:** guang rong, Lauri Ahonen.

**Visualization:** guang rong, Lauri Ahonen.

**Writing – original draft:** guang rong, Lauri Ahonen.

**Writing – review & editing:** guang rong, Lauri Ahonen, Gerit Pfuhl, Benjamin Ultan Cowley.

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
