## [Decision Letter · Decision Letter 0]

14 Oct 2024

PGPH-D-24-01671

Death of backcountry winter-sports practitioners in avalanches -- a systematic review and meta-analysis of proportion of causes of avalanche death

Dear Dr. rong,

Thank you for submitting your manuscript to PLOS Global Public Health. After careful consideration, we feel that it has merit but does not fully meet PLOS Global Public Health’s publication criteria as it currently stands. Therefore, we invite you to submit a revised version of the manuscript that addresses the points raised during the review process.

Please revise your manuscript with specific care to address the concerns of reviewer 3 who raised concerns regarding the conciseness of your reporting and ability for the reader to understand your study design.

We look forward to receiving your revised manuscript.

Kind regards,

Jennifer Tucker, PhD

Staff Editor

Journal Requirements:

**Please only choose the relevant sentences from below**

1. Please clarify all sources of funding (financial or material support) for your study. List the grants (with grant number) or organizations (with url) that supported your study, including funding received from your institution. 

2. State the initials, alongside each funding source, of each author to receive each grant.

3. State what role the funders took in the study. If the funders had no role in your study, please state: “The funders had no role in study design, data collection and analysis, decision to publish, or preparation of the manuscript.”

4. If any authors received a salary from any of your funders, please state which authors and which funders.

3. We ask that a manuscript source file is provided at Revision. Please upload your manuscript file as a .doc, .docx, .rtf or .tex.

4. Please provide separate figure files in .tif or .eps format.

5. We noticed that you used "not shown” in the manuscript. We do not allow these references, as the PLOS data access policy requires that all data be either published with the manuscript or made available in a publicly accessible database. Please amend the supplementary material to include the referenced data or remove the references.

6. We notice that your figures 1 and 2 are included in the manuscript file. Please remove them as there are already uploaded in the File Inventory.

7. As required by our policy on Data Availability, please ensure your manuscript or supplementary information includes the following: 

Additional Editor Comments (if provided):

Reviewers' comments:

Reviewer's Responses to Questions

**Comments to the Author**

1. Does this manuscript meet PLOS Global Public Health’s publication criteria? Is the manuscript technically sound, and do the data support the conclusions? The manuscript must describe methodologically and ethically rigorous research with conclusions that are appropriately drawn based on the data presented.

Reviewer #1: Yes

Reviewer #2: Yes

Reviewer #3: Partly

2. Has the statistical analysis been performed appropriately and rigorously?

Reviewer #1: Yes

Reviewer #2: No

Reviewer #3: I don't know

3. Have the authors made all data underlying the findings in their manuscript fully available (please refer to the Data Availability Statement at the start of the manuscript PDF file)?

Reviewer #1: Yes

Reviewer #2: Yes

Reviewer #3: Yes

4. Is the manuscript presented in an intelligible fashion and written in standard English?

Reviewer #1: Yes

Reviewer #2: Yes

Reviewer #3: No

5. Review Comments to the Author

Reviewer #1: Systematic review and meta-analysis assessing the reasons of death of back-country practitioners in avalanches

Very long and thorough manuscript with plenty of information. Overall, important statements and good addition to what is known.

In the abstract 'PCAD' - Proportions of causes of avalanche death - should be correctly described. It seems that the journal site has not formated the original abstract correctly, see first page of the .pdf file, e.g. 'Between-study heterogeneity was assessed jointly by *I*^2^ and 95% prediction interval of pooled

estimates. Meanwhile, we performed a narrative review to supplement the meta-synthesis' and in the conclusions 'A sample size $>$ 75 is needed to estimate the proportion of hypothermia'

Several references to not correspond to the journal style and relevant data eg journal is missing, see for example ref 3, 10, 11 and many others

Reviewer #2: I see a significant issue with combining proportion estimates for binary outcomes in subgroup analyses, as these estimates may not be directly comparable without proper normalization. Therefore, the statistical analysis is only partially conducted appropriately and rigorously.

The article is highly detailed and represents the most comprehensive review to date on the subject, making a significant and valuable contribution to the field.

Please see the attachment for details.

Reviewer #3: The paper is dense and a very tough read if a general readership is desired. It is extremely long (55 pages of regular text), it reads more like a thesis/dissertation rather than a focused journal manuscript. For me, it was a tough read, and I mostly just commented on the things I know.

The authors studied major causes of avalanche deaths. Trauma, asphyxia, hypothermia – not surprising asphyxia is the cause, well known to many international avalanche communities (American, Canadian, European, Japan) as most victims are buried for lengthy periods of time for deprivation of oxygen. Some have trauma in combination, it is occasionally tough to differentiate with that alone from asphyxia. Asphyxia as a major cause is not at all surprising nor a big new find to me.

The authors did a mathematical meta-analysis approach. I have no expertise on this approach, The authors seemed knowledgeable and their setup of biased verses unbiased looks good, but again I am no expert and won’t comment in this review. I hope that there are other reviewers that can comment on these aspects. The flowchart summarizes their approach well but I am not sure it could be appealing to general Plos Global Public Health Review readers.

1415 reports but why a much number for meta-analysis?

I am skeptical on the quality of some of the databases (electronically) used to search for avalanche death data. The authors should focus on official avalanche data from things like the USA Westwide Network and reported avalanche incident reports, and similar official/operation databases. Things like Academic Search Complete, SPORTDIscus, etc. may not have a true picture on whether it was officially a true avalanche scientifically. The authors provide little information on the geography of avalanche incidents. I am pleased they looked for non-English sources, but comprehensive good avalanche incident data is critical.

For at least the United States (can be different in some international countries), it is well-known that back-country skiers have a much different aspect on avalanche death issues than those in regular ski-areas (pretty much non-existent)

The section on Weather and Snow doesn’t consider detailed meteorological factors within a climate timeframe that could be important on avalanche fatalities.

I wouldn’t expect terrain to be a big factor in fatalities.

On changes of factors through time on fatalities, I would like to see some clear visual(s) that show this over the years, but with the sketchy nature of most avalanche fatality data, especially in some remote areas in some cases, its hard to do.

6. PLOS authors have the option to publish the peer review history of their article (what does this mean?). If published, this will include your full peer review and any attached files.

**Do you want your identity to be public for this peer review?** For information about this choice, including consent withdrawal, please see our Privacy Policy.

Reviewer #1: **Yes: **Peter Paal

Reviewer #2: **Yes: **Markus Falk

Reviewer #3: No

---

## [Decision Letter · Decision Letter 1]

4 Mar 2025

PGPH-D-24-01671R1

Death of backcountry winter-sports practitioners in avalanches -- a systematic review and meta-analysis of proportion of causes of avalanche deathCopyTranslateCopyTranslate

Dear Dr. rong,

Thank you for submitting your manuscript to PLOS Global Public Health. After careful consideration, we feel that it has merit but does not fully meet PLOS Global Public Health’s publication criteria as it currently stands. Therefore, we invite you to submit a revised version of the manuscript that addresses the points raised during the review process.

Reviewer 2 has identified an inconsistency in your discussion section which requires clarification and addressing before we can consider your manuscript for publication. 

We look forward to receiving your revised manuscript.

Kind regards,

Jennifer Tucker, PhD

Staff Editor

Journal Requirements:

Additional Editor Comments (if provided):

Reviewers' comments:

Reviewer's Responses to Questions

**Comments to the Author**

1. If the authors have adequately addressed your comments raised in a previous round of review and you feel that this manuscript is now acceptable for publication, you may indicate that here to bypass the “Comments to the Author” section, enter your conflict of interest statement in the “Confidential to Editor” section, and submit your "Accept" recommendation.

Reviewer #1: All comments have been addressed

Reviewer #2: All comments have been addressed

2. Does this manuscript meet PLOS Global Public Health’s publication criteria? Is the manuscript technically sound, and do the data support the conclusions? The manuscript must describe methodologically and ethically rigorous research with conclusions that are appropriately drawn based on the data presented.

Reviewer #1: Yes

Reviewer #2: Yes

3. Has the statistical analysis been performed appropriately and rigorously?

Reviewer #1: Yes

Reviewer #2: Yes

4. Have the authors made all data underlying the findings in their manuscript fully available (please refer to the Data Availability Statement at the start of the manuscript PDF file)?

Reviewer #1: Yes

Reviewer #2: Yes

5. Is the manuscript presented in an intelligible fashion and written in standard English?

Reviewer #1: Yes

Reviewer #2: Yes

6. Review Comments to the Author

Reviewer #1: Well revised manuscript.

Some counting mistake has occurred: In line 889 the authors mention four results of this review: After the third, in line 892, the go on with Fifth line 894 and sixth line 895 without mentioning the fourth. Please correct

Reviewer #2: 1) please add the file MetaData2024.xlsx to the repository https://osf.io/ygza4/.

2) In line 753 on page 35/66 there are two question marks (??) in the text.

7. PLOS authors have the option to publish the peer review history of their article (what does this mean?). If published, this will include your full peer review and any attached files.

**Do you want your identity to be public for this peer review?** For information about this choice, including consent withdrawal, please see our Privacy Policy.

Reviewer #1: **Yes: **Peter Paal

Reviewer #2: **Yes: **Markus Falk

---

## [Decision Letter · Decision Letter 2]

4 Apr 2025

Death of backcountry winter-sports practitioners in avalanches -- a systematic review and meta-analysis of proportion of causes of avalanche deathCopyTranslateCopyTranslateCopyTranslateCopyTranslateCopyTranslate

PGPH-D-24-01671R2

Dear Dr. rong,

We are pleased to inform you that your manuscript 'Death of backcountry winter-sports practitioners in avalanches -- a systematic review and meta-analysis of proportion of causes of avalanche deathCopyTranslateCopyTranslateCopyTranslateCopyTranslateCopyTranslate' has been provisionally accepted for publication in PLOS Global Public Health.

Please ensure that you have removed the words "CopyTranslateCopyTranslateCopyTranslateCopyTranslateCopyTranslate" from the title of your manuscript as you go through the final formatting checks.

Best regards,

Julia Robinson

Executive Editor

Reviewer Comments (if any, and for reference):

Reviewer's Responses to Questions

**Comments to the Author**

1. If the authors have adequately addressed your comments raised in a previous round of review and you feel that this manuscript is now acceptable for publication, you may indicate that here to bypass the “Comments to the Author” section, enter your conflict of interest statement in the “Confidential to Editor” section, and submit your "Accept" recommendation.

Reviewer #1: All comments have been addressed

Reviewer #2: All comments have been addressed

2. Does this manuscript meet PLOS Global Public Health’s publication criteria? Is the manuscript technically sound, and do the data support the conclusions? The manuscript must describe methodologically and ethically rigorous research with conclusions that are appropriately drawn based on the data presented.

Reviewer #1: Yes

Reviewer #2: Yes

3. Has the statistical analysis been performed appropriately and rigorously?

Reviewer #1: Yes

Reviewer #2: Yes

4. Have the authors made all data underlying the findings in their manuscript fully available (please refer to the Data Availability Statement at the start of the manuscript PDF file)?

Reviewer #1: Yes

Reviewer #2: Yes

5. Is the manuscript presented in an intelligible fashion and written in standard English?

Reviewer #1: Yes

Reviewer #2: Yes

6. Review Comments to the Author

Reviewer #1: Well revised manuscript, thank you

Reviewer #2: (No Response)

7. PLOS authors have the option to publish the peer review history of their article (what does this mean?). If published, this will include your full peer review and any attached files.

**Do you want your identity to be public for this peer review?** For information about this choice, including consent withdrawal, please see our Privacy Policy.

Reviewer #1: **Yes: **Peter Paal

Reviewer #2: **Yes: **Markus Falk
